# *Mycobacterium tuberculosis* SecA2-dependent activation of host Rig-I/MAVs signaling is not conserved in *Mycobacterium marinum*

**Lindsay G. Serene**[1,2]*, **Kylie Webber**[1,2], **Patricia A. Champion**[1,2], **Jeffrey S. Schorey**[1,2]*

**1** Department of Biological Sciences, University of Notre Dame, Notre Dame, IN, United States of America,
**2** Eck Institute for Global Health, University of Notre Dame, Notre Dame, IN, United States of America

* lserene@nd.edu, lougrace@protonmail.ch (LGS); schorey.1@nd.edu (JSS)

**Data Availability Statement:** All data and R code are available at https://github.com/Lindsayeg/MMAR_SecA2_ManuscriptData_2023.git.

**Funding:** This work was supported by the Eck Institute for Global Health Graduate Student

## Abstract

Retinoic acid inducible gene I (Rig-I) is a cytosolic pattern recognition receptor canonically described for its important role in sensing viral RNAs. Increasingly, bacterially-derived RNA from intracellular bacteria such as *Mycobacterium tuberculosis*, have been shown to activate the same host Rig-I/Mitochondrial antiviral sensing protein (MAVS) signaling pathway to drive a type-I interferon response that contributes to bacterial pathogenesis *in vivo*. In *M. tuberculosis*, this response is mediated by the protein secretion system SecA2, but little is known about whether this process is conserved in other pathogenic mycobacteria or the mechanism by which these nucleic acids gain access to the host cytoplasm. Because the *M. tuberculosis* and *M. marinum* SecA2 protein secretion systems share a high degree of genetic and functional conservation, we hypothesized that Rig-I/MAVS activation and subsequent induction of IFN-β secretion by host macrophages will also be conserved between these two mycobacterial species. To test this, we generated a Δ*secA2 M. marinum* strain along with complementation strains expressing either the *M. marinum* or *M. tuberculosis secA2* genes. Our results suggest that the Δ*secA2* strain has a growth defect *in vitro* but not in host macrophages. These intracellular growth curves also suggested that the calculation applied to estimate the number of bacteria added to macrophage monolayers in infection assays underestimates bacterial inputs for the Δ*secA2* strain. Therefore, to better examine secreted IFN-β levels when bacterial infection levels are equal across strains we plated bacterial CFUs at 2hpi alongside our ELISA based infections. This enabled us to normalize secreted levels of IFN-β to a standard number of bacteria. Applying this approach to both WT and MAVS[-/-] bone marrow derived macrophages we observed equal or higher levels of secreted IFN-β from macrophages infected with the Δ*secA2 M. marinum* strain as compared to WT. Together our findings suggest that activation of host Rig-I/MAVS cytosolic sensors and subsequent induction of IFN-β response in a SecA2-dependent manner is not conserved in *M. marinum* under the conditions tested.

Fellowship, awarded to LGS. PAC is supported by the National Institutes of Health under award numbers AI156229, AI106872, AI149147, and AI149235. The content of this article is solely our responsibility and does not necessarily represent the official views of the National Institutes of Health. The funders had no role in study design, data collection and analysis, decision to publish, or preparation of the manuscript.

**Competing interests:** The authors have declared that no competing interests exist.

## Introduction

While originally characterized for their ability to interfere with viral replication, type I interferons are increasingly recognized as important contributors in host-pathogen interactions involving intracellular bacteria. Studies in *Francisella tularensis* [1–4], *Salmonella enterica* serovar Typhimurium [5–7], *Listeria monocytogenes* [8, 9] and *Mycobacterium tuberculosis* [10, 11] have shown that these bacteria can specifically induce host type I interferons and that induction of this signaling pathway can directly impact host:bacteria interactions *in vivo*. For example, multiple studies have demonstrated that patients with active tuberculosis disease have blood transcriptomic signatures dominated by an upregulation of type I interferon inducible genes, including interferon-β (IFN-β). These signatures correlated with lung pathology by X-ray and were observed to resolve upon successful treatment of the disease [12–19].

In *M. tuberculosis*, there are multiple mechanisms by which IFN-β is induced following infection [20–24], including through the activation of host cytosolic RNA sensors canonically described for their role in driving the type I interferon response to viruses [11, 25, 26]. Activation of these host RNA sensors has been shown to be driven by the release of bacterially derived RNAs into the host cytosol [11]. Once within the cytosol, these immunostimulatory RNAs are recognized by retinoic acid-inducible gene I (RIG-I), causing a conformational change in this protein which enables it to interact with a signal transducing adaptor protein, mitochondrial antiviral-signaling protein (MAVS). Through a series of phosphorylation-dependent steps, this signal is ultimately relayed to transcription factors such as interferon regulatory factor 7 (Irf7), which translocate into the nucleus to drive IFN-β transcription [27]. The mechanism by which these *M. tuberculosis*-derived RNAs gain access to the host cytosol is unclear, but it is known to be dependent, in part, on a functional SecA2-protein secretion system.

The SecA2 protein secretion system is a non-essential, auxiliary protein secretion system present in all mycobacteria and some Gram-positive bacteria [28]. In addition to its role in contributing to the release of (myco)bacterial RNAs, it is also responsible for transporting a subset of unfolded pre-proteins from the bacterium's cytoplasm to the host cytosol. Once across the bacterial cell wall, these proteins can fold into their mature structures and enact specific effector functions. An example of one such protein is PknG, a serine threonine kinase which contributes to phagosome maturation arrest [29, 30] as well as metabolism and redox homeostasis [31–33].

*M. marinum*, a close genetic relative of *M. tuberculosis*, is a common model organism utilized to explore the molecular mechanisms of *M. tuberculosis* virulence due to its less restrictive containment measures, faster doubling time, and shared mechanisms of survival and persistence in host macrophages [34–36]. While *M. marinum* has been used to characterize the genetic underpinnings of other protein secretions systems, including the 6-kDa early secretory antigenic target (ESAT-6) secretory system-1 (ESX-1) [37–40], which is also required for *M. tuberculosis* pathogenesis [41, 42], it has not yet been established as a model for SecA2-dependent IFN-β activation.

*M. tuberculosis* SecA2 shares a high degree of genetic (83.28% identity at the nucleotide and 88.34% at the amino acid level) and functional conservation with *M. marinum*. For example, in both *M. tuberculosis* and *M. marinum* SecA2 is required for the export of PknG [43–45], host phagosomal maturation arrest [30, 43, 46], and attenuation of the bacterium *in vivo* [29, 46]. For these reasons, we sought to determine whether SecA2-dependent activation of host Rig-I/MAVS cytosolic sensors and subsequent induction of IFN-β is conserved between these two mycobacterial species. We hypothesized that based on their shared genetic and functional conservation this pathogenesis mechanism would be conserved, providing a biosafety level 2

model organism that could be utilized to determine the molecular mechanism driving this important pathogenesis pathway.

To test this hypothesis, we generated a Δ*secA2 M. marinum* strain as well as *M. marinum* complementation and *M. tuberculosis* cross complementation constructs. Using these strains, we examined the impact *secA2* has on bacterial growth and virulence under a variety of different *in vitro* and *ex vivo* conditions. To specifically address our primary aim of examining whether the SecA2 protein secretion system contributes to an RNA driven, IFN-β mediated pathogenesis mechanism in *M. marinum* as it does in *M. tuberculosis*, we infected WT and MAVS[-/-] BMDMs with our generated strains and examined the macrophage secretome for secreted levels of IFN-β 24-hours post infection.

## Methods

### Bacteria strain maintenance

*Mycobacterium marinum* M bacterial strains were grown on Middlebrook's 7H11 (Sigma-Aldrich) agar plates supplemented with 10% OADC [oleic acid (Fischer Scientific), dextrose (Sigma and VWR), albumin (Sigma Aldrich), and catalase(Fischer)] and 0.5% glycerol (VWR, Radnor, PA) at 30˚C or in Middlebrook's 7H9 media (Sigma-Aldrich, St. Louis, MO) supplemented with 10% OADC and 0.2% tyloxapol (Chem-Impex Int'l Inc., Wood Dale, IL) at 30˚C. Hygromycin (EMP Millipore) and Kanamycin (IBI Scientific, Peosta, IA) were added to final concentrations of 50 μg/mL and 20 μg/mL, respectively, where noted. *Escherichia coli* was grown on Luria-Bertani (LB) agar or in LB broth at 37˚C. Kanamycin, Hygromycin, and Ampicillin (VWR) were added to final concentrations of 50 μg/mL, 200 μg/mL, and 200 μg/mL when stated. Please see S1 Table for a complete list of bacterial strains used in this study.

### Mice

Wild type (Jackson Laboratories, Bar Harbor, ME) and MAVS[-/-] [Dr. Stanley Perlman, [11, 47]] C57BL/6 mice were bred and housed at the Freimann Life Science Center at the University of Notre Dame [Animal Welfare Assurance (#A3093-01)]. All animals were cared for in accordance with the Association for Assessment and Accreditation of Laboratory Animal Care International under pathogen-free conditions, and all animal experiments were approved by the University of Notre Dame's Institutional Animal Care and Use Committee.

### Isolation of BMDMs

Bone marrow derived macrophages were isolated from the femurs of 6–8 week old C57BL/6 mice (The Jackson Laboratory) following euthanasia by cervical dislocation. Briefly, bone marrow was collected into a 50 mL conical tube by flushing femur with 1X phosphate buffered saline (PBS; Hyclone, Logan, Utah) + 1% penicillin/streptomycin and pelleted by centrifugation at 1000 rpm for 5 minutes at 4˚C. Cells were washed once in 1X PBS with gentle agitation before being pelleted again by centrifugation at 1000 rpm for 5 minutes at 4˚C. The pellet was resuspended in sterile ACK solution (2.5 mL/mouse; 150 mM $NH_4Cl$, 1mM $KHCO_3$, and 0.1 mM EDTA filter sterilized through a 0.2 μM filter) and incubated for 7–10 minutes to lyse red blood cells. The remaining cell suspension was then pelleted, washed twice in 1xPBS, resuspended in BMDM media, and transferred to 100 x 15 mm petri dishes at a density of $1x10^7$ cells/dish in 10 mL media. Cells were grown for seven days, with one media change on day 3, and then stocked down in DMEM + 20% HI-FBS + 10% DMSO at a density of $1x10^6$ cells/mL.

## Mammalian cell culture

Bone marrow derived macrophages (BMDMs) were maintained in Dulbecco's Modified Eagle Medium (DMEM; Sigma-Aldrich and Sigma) supplemented with 20% L929 cell conditioned media, 10% heat-inactivated fetal bovine serum (HI-FBS; Hyclone and Gibco), and 100 U/mL penicillin/streptomycin (Cytiva, South Logan, UT) at 37°C with 5% $CO_2$. L929 fibroblasts were maintained in DMEM supplemented with 10% HI-FBS and 100 U/mL penicillin/streptomycin (Cytiva; unless otherwise noted) at 37°C with 5% $CO_2$.

## L929 supplement

L929 fibroblasts were rapidly thawed in a 37°C water bath before being washed once in DMEM and 10% HI-FBS and transferred to a T-25 tissue culture treated flask (Sarstedt, Nümbrecht, Germany) in 10 mL DMEM and 10% HI-FBS with 100 U/mL penicillin/streptomycin. At about 70% confluency, the cells were detached in 0.05% trypsin-EDTA (Gibco, Grand Island, NY) and bumped up to T-75 in 20 mL DMEM and 10% HI-FBS with 100 U/mL penicillin/streptomycin. This process was repeated, bumping the cells into a T-125 flask with 40 mL of the same media. At 70% confluency, the cells were again detached and then split equally between five flasks of the same size. This was repeated once more with each of the five flasks seeding an additional five flasks for a total of 25, T-125 flasks with 40 mL DMEM and 10% HI-FBS (no antibiotics). The cells were then allowed to incubate for 10 days at 37°C and 5% $CO_2$, before the supernatants were harvested, passed through a 0.2 μM filter, and stored in 50mL aliquots at -80°C until further use.

## Generation of electrocompetent *M. marinum*

Electrocompetent *M. marinum* were generated from single colonies grown in 5mL 7H9 + 0.1% tween 80 for 3–5 days before being expanded to 25 mLs in the same media. Twenty-four to forty-eight hours later, 2 mL 2M Glycine was added and the cells incubated for twenty-four hours. Cells were then pelleted at 3,500 rpm for 5 minutes, washed three times in comp. cell buffer (distilled water containing 10% glycerol and 0.1% tween 80),resuspended in 5mL comp. cell buffer (1/5th the original culture volume), and frozen in 500 μl aliquots at -80C until further use.

## Generation of the *ΔsecA2* knockout strain

The *ΔsecA2* strain was generated using a previously established allelic exchange approach [48]. Briefly, a 1.5 kb region of DNA upstream and downstream of *secA2 M. marinum* M genome (NC_010612.1) was amplified by PCR using Phusion polymerase (a complete list of primers and plasmids can be found in S2 and S3 Tables, respectively). Amplicons were introduced into DPNI treated p2NIL vector (Addgene, #20188) by three-part fast cloning [49], transformed into chemically competent DH5α *E. coli*, and plated on LB agar containing 50 μg/mL kanamycin. Single colonies were picked and grown overnight in LB broth with 50 μg/mL Kanamycin before plasmids were isolated using a QIAprep Spin Mini Kit (Qiagen, Germantown, MD), confirmed by restriction enzyme digest, and quantified by A260 using a Nanodrop 2000 Spectrophotometer (ThermoScientific, Rockford, IL). Verified p2NIL/*ΔsecA2* plasmids were digested with PacI (NEB; 37°C for 1 hour, 65°C for 20 minutes), dephosphorylated with antarctic phosphatase (NEB; 37°C for 30 minutes, 80°C for 2 minutes), and ligated with the pGOAL (Addgene, #20190) gene marker cassette by overnight ligation at 4°C with T4 DNA Ligase (NEB). Overnight ligations were transformed into chemically competent DH5α *E. coli* and selected for on LB agar containing 50 μg/mL Kanamycin and 60 μg/mL X-Gal (5 Prime,

Gaithersburg, Maryland). Blue colonies were picked and grown overnight in LB broth containing 50 μg/mL kanamycin before plasmids were isolated, confirmed by restriction enzyme digestion, and quantified as before.

Verified p2NIL/Δ*secA2* GOAL plasmids were transformed into electrocompetent *M. marinum* M bacteria at room temperature. One 500μl stock of WT electrocompetent cells were thawed at room temperature, pelleted at 13,000rpm for 1 minute, washed three times with glycerol wash buffer (10% glycerol with 0.1% Tween 80), resuspended in 500μl of the same buffer, and placed between the metal plates of a 2 mm electroporation cuvette (VWR). Three micrograms of p2NIL/Δ*secA2* GOAL was UV irradiated (exposed to 1000 μJ/cm2 in a CL-1000 Ultraviolet Crosslinker) was added to the electroporation cuvette. The sample was then electroporated on a BioRad GenePulser Xcell under the following conditions: 2500V voltage, 25μF capacitance, and 1000Ω resistance. Bacteria were allowed to recover overnight in 7H9 + 0.1% Tween80 before being plated on 7H11 agar containing 50 μg/mL Kanamycin, and 60 μg/mL X-Gal and incubated at 30°C for two weeks.

Single blue colonies (merodiploids) were picked and cultured in 7H9 for five days before being counter selected for double crossover events on 7H11 agar containing 50 μg/mL Kanamycin, 60 μg/mL X-Gal, and 2% sucrose and incubated at 30°C for 14 days. White colonies were picked, cultured five days in 7H9 + 0.1% tween 80 and genotyped. Deletion of *secA2* was confirmed by Sanger sequencing using primers secA2-X and secA2-Y by the Genomics core at the University of Notre Dame and the resulting sequences analyzed by FinchTV (version 1.5.0). Additionally, strains were checked for the deletion of *secA2* using a rapid genotyping process previously described [50].

## Generation of complementation strains

*M. marinum* (MMAR_2698) complementation and *M. tuberculosis* (Rv1821) cross complementation strains were generated using an integrative vector (pMV306H) under the control of the mycobacterial optimal promoter (MOP; George et al., 1995). To begin, *secA2* was amplified from either *M. marinum* M or *M. tuberculosis* CDC1551 genomic DNA with primers MMAR_2698 P1 and MMAR_2698 P2 or Rv1821 P1 and Rv1821 P1, respectively, using Phusion High Fidelity polymerase with HF buffer and 3% DMSO (NEB) under the following cycling conditions: 98°C, 30 seconds; 98°C, 10 seconds, 72°C, 30 seconds, 72°C, 30 seconds for 30 cycles; 72°C, 5 minutes. PCR amplicons were checked for size on a 1% agarose gel and imaged using a BioRad GelDoc 1000.

Verified amplicons were combined with the pMV306H integrative vector backbone at a ratio of 1:1 v/v and allowed to ligate overnight at 4°C with T4 DNA ligase (NEB). Ligation products were subsequently transformed into chemically competent DH5α *E.coli*. Following a 1 hour recovery/out-growth period, transformants were plated onto LB agar containing 200 μg/mL hygromycin. The next day, single colonies were inoculated into LB broth with 200 μg/mL hygromycin and allowed to grow overnight before generated plasmids were purified by mini-prep using a QIAprep Spin Miniprep Kit. Plasmids were confirmed by restriction enzyme digest and concentrations quantified by A260 using a Nanodrop 2000 Spectrophotometer.

Verified plasmids were electroporated into electrocompetent Δ*secA2 M. marinum* at room temperature. Electrocompetent bacteria, prepared the same as previously described, were combined with 500 ng of verified plasmid (no UV-irradiation) between the metal plates of a 2mm electroporation cuvette. The samples were electroporated on a BioRad GenePulser Xcell under the following conditions: 2500V voltage, 25μF capacitance, and 1000Ω resistance. Bacteria were allowed to recover overnight in 7H9 + 0.1% Tween80 before being plated on 7H11 agar

containing 200 μg/mL Hygromycin and incubated at 30˚C for two weeks. Single colonies were picked and cultured in 7H9 for five days before being checked for the complementation using a rapid genotyping process [50].

## Western blot

Bacteria were pelleted from mid-log phase cultures at 3,500 rpm for 10 minutes at 4˚C. Following centrifugation the supernatants were poured off and the pellets stored at -80˚C. These pellets were subsequently thawed on ice, spun down at 3,500 rpm for 10 minutes at 4˚C, and any remaining supernatant removed. To generate whole cell lysates, the bacteria were resuspended in 1/10th of the original culture volume of ice cold 1 x Dulbecco' phosphate buffered saline (DPBS; Hyclone) containing 1 mM phenylmethylsulfonyl fluoride (PMSF) and added to a 2.0 mL O-ring screw cap micro tube (Sarstedt) with 0.1 mm zirconia beads (RPI, Mount Prospect, IL). Bacteria were mechanically lysed by three, thirty-second pulses on a Mini-Beadbeater (BioSpec). The resulting membranes were pelleted by centrifugation at 13,000 rpm for 15 minutes, and the supernatant was transferred to a new tube. The protein concentration for each supernatant was determined using a Pierce BCA Protein Assay Kit (ThermoFischer Scientific).

Mammalian cells were lysed directly in the twenty-four well plates, following the removal of culture supernatants. Monolayers were lysed in 75 μl, ice-cold RIPA lysis buffer (150mM sodium chloride; 50mM Tris-HCl, pH 8.0; 1% Triton X100; 0.5% sodium deoxycholate; 0.1% SDS) containing 1 mM PMSF added directly before use. The resulting whole cell lysates were centrifuged a 12,000 rpm for 5 minutes before quantifying the protein concentration using a Pierce BCA Protein Assay Kit (ThermoFischer Scientific)

Equal amounts of protein [60 μg for bacterial whole cell lysates (WCL); 15 μg for macrophage WCLs) were added to 5X SDS loading dye [250 mM Tris-HCl, pH 6.8 (Fischer Scientific, Fair Lawn, NJ); 10% sodium dodecyl sulfate (SDS; Amresco, Solon, OH); 30% glycerol (VWR); 0.02% Bromophenol blue (Fischer Scientific); and 5% β-Mercaptoethanol (Sigma-Aldrich)] and topped up to a final volume of 40 μl with 1X DPBS before being boiled for 10 minutes and then loaded into lanes of an 8% SDS-PAGE gel. Following electrophoresis, the separated proteins were transferred onto a methanol activated 0.45 μM PVDF membrane (Millipore) at 70V for 120 minutes at 4˚C in Transfer Buffer (192 mM Glycine, 25 mM Tris Base, and 10% Methanol). Membranes were blocked in 5% non-fat milk diluted in TBS-T (20 mM Tris Base, 150 mM NaCl, and 0.5% tween 20; pH = 7.6) for one hour at room temperature with gentle agitation. Membranes were then incubated with primary antibody diluted in blocking buffer overnight at 4˚C, rocking. The following day, membranes were washed five times in TBS-T for five-minutes before incubation with an HRP-conjugated secondary antibody diluted in blocking buffer at room temperature for 3 hours. Non-specifically and unbound secondary antibody was removed by washing the membrane five times in TBS-T for 5 minutes. Membranes were coated in SuperSignal West PicoPlus Chemiluminescent or Femto Maximum Sensitivity Substrate (ThermoScientific), exposed to BioBlue-MR Autoradiography film (Alkali Scientific), and imaged on a Konica SRX-101A developer. Please see S4 Table for a list of antibodies utilized in this study.

## *In vitro* growth curve

*M. marinum* bacterial strains were grown to exponential phase at 30˚C in 5 mLs of 7H9 + 10% OADC + 0.2% tyloxapol in 25 mL Erlenmeyer flasks, shaking at 150 rpm. These were then subcultured into triplicate 125 mL Erlenmeyer flasks with 25 mLs of the same media at an optical

density of 0.05 using the following equation:

$$Volume\ active\ culture\ add\ subculture\ (mL) = \frac{Desired\ OD600\ x\ Desired\ subculture\ volume\ (mL)}{OD600\ active\ bacterial\ culture}$$

Initial OD readings were measured on an Eppendorf BioPhotometer at the time of subculturing and then once every twenty-four hours for the duration of the experiment. Following data collection, growth curve data were summarized using the R package Growthcurver [51], which fits a logarithmic model [K / (1 + ((K—N0) / N0) * exp(-r * t)] to each curve. Estimated growth rates (r) were compared across bacterial strains to identify changes in growth rates.

## SDS sensitivity assay

Previous work in an *M. marinum* M Δ*secA2* strain showed increased sensitivity to sodium-dodecyl sulfate (SDS), indicating a cell wall defect (Watkins et al., 2012). We examined our Δ*secA2* strain for this same phenotype. WT, Δ*secA2*, and Δ*secA2*/p*secA2*$_{MM}$ were grown in 5 mL 7H9 + 10% OADC + 0.2% tyloxapol for 3–5 days before being bumped up to 25 mLs. Once at mid-log phase, these cultures were subcultured into 5 mL of the same media, 7H9 + 10% OADC + 0.2% tyloxapol + 0.05–0.2% SDS at an OD600 of 0.8, and allowed to grow 24-hours at 30˚C, shaking at 150 rpm. Twenty-four hours later, 50μl of each culture was serially diluted in PBS + 0.2% tyloxapol and plated onto 7H11 + 10% OADC agar plates in technical triplicate. Plates were incubated at 32˚C with 5% $CO_2$ for one week before enumeration.

## Macrophage infections

Bone marrow derived macrophages (BMDMs) were cultured in BMDM media for seven days, with media changed at 24 and 96 hours. On day seven, macrophage monolayers were washed once with sterile, 1X phosphate buffered saline (Hyclone, Logan, Utah) and dissociated with 0.05% Trypsin EDTA (Gibco). Trypsin was inactivated with the addition of BMDM media and the cells were pelleted by centrifugation at 250 x g for 5 minutes. Cells were then resuspended in fresh BMDM media, enumerated on a 0.1 mm Brightline hemocytometer (Hausser Scientific, Horsham, PA), seeded into wells of 24 or 48-well tissue culture plates and incubated overnight at 37˚C with 5% $CO_2$. For intracellular growth assays, macrophages were seeded into wells of a 48-well tissue culture plate at a density of 1 x $10^5$ cells in 500 μL BMDM media, and at a density of 5 x $10^5$ cells in 1 mL BMDM media in 24-well plates for ELISA based assays. The next morning, macrophages were infected with a single-cell suspension of log-phase bacteria at a multiplicity of infection of 0.2 (1 bacteria for every 5 macrophages) for intracellular growth assays or 1 for ELISA assays.

Prior to infection, bacteria were collected by centrifugation at 3,500 rpm for 10 minutes, washed once with 1X PBS, and then resuspended in 1 mL 1X PBS. Bacterial suspensions were incubated at room temperature for 30 minutes to allow clumps to settle. The top 700 μl of suspension was transferred to a fresh tube and the optical density (OD600) measured. The volume of bacteria needed to infect at a standard MOI for each strain was calculated based on the assumption that at an $OD_{600}$ of 1 there are 7.7 x $10^7$ CFU/mL using the following equation:

$$Volume\ bacteria = \frac{Desired\ number\ of\ bacteria\ (CFU/mL)}{(OD_{600}\ of\ active\ culture)\ x\ (7.7 x 10^7\ CFU/mL)}$$

Macrophages and bacteria were incubated together for 2 hours at 32˚C with 5% CO2 before the supernatant was discarded and monolayers washed three times with 1X DPBS to remove extracellular bacteria. For intracellular growth assays, the 48 and 96 hour time points were further incubated with BMDM media containing 100 ug/mL Gentamycin (VWR) until 6 hours

post infection, at which point the supernatant was removed and media replaced with BMDM lacking antibiotics. For intracellular and IFN-β based assays, following the third PBS wash, the 2 hour time point was lysed in 250 (intracellular) or 500 (ELISA) μl sterile water + 0.2% tyloxapol, diluted in 1X DPBS + 0.2% tyloxapol and plated onto 7H11 + 10% OADC agar plates. Plates were incubated for 1 week at 32˚C with 5% $CO_2$ before being counted. At 48 and 96 hours post infection, the culture supernatant was removed prior to lysing, diluting, and plating the macrophage monolayer as described for the 2 hour timepoint. Additionally, media was replaced on the 96 hour timepoint at 48 hours post infection. Plates were incubated at 32˚C for one week before colonies were enumerated. Intracellular growth was assessed by fitting a linear regression model to colony forming units counted over time. Estimated growth rates (r) were compared using a one-way ANOVA followed by Dunnett's multiple comparison test.

## ELISA

Macrophages were infected as described under "Bacterial Growth within BMDMs" with a couple exceptions. Macrophage monolayers were infected at an MOI of 1, unless otherwise stated, and following the 2 hour infection, extracellular bacteria were removed by washing macrophage monolayers three times with 1XDPBS (Cytiva/HyClone). Following the third wash, fresh BMDM media (without antibiotics) was added and the plates were placed at 32˚C with 5% $CO_2$. Secreted levels of IFN-β were measured from BMDM supernatants twenty-four hours following the start of the infection following the Sandwich ELISA protocol published by BioLegend. Anti-IFN-β capture antibody (BioLegend, SanDiego, CA) was diluted 1:200 in Carbonate Coating Buffer [0.84% w/v $NaHCO_3$ (Sigma) and 0.356% w/v $Na_2CO_3$ (Sigma) dissolved in $ddH_2O$, pH = 9.5], while the detection antibody (Biotin anti-mouse IFN-β; BioLegend) and Avidin-HRP (BioLegend) were diluted 1:200 and 1:500, respectively, in blocking buffer. Absorbance was measured at a wavelength of 405 nm and secreted levels of IFN-β were calculated from blank corrected data based on absorbance of the IFN-β standard at known concentrations.

## IFN-β normalization

Secreted levels of IFN-β were normalized to 10,000 bacteria using the equation a/∑b, where "a" is the amount of IFN-β measured for each technical replicate and "∑b" is the sum of CFUs counted from technical replicates originating from matching wells of the same biological experiment. This number was then multiplied by 10,000 to provide IFN-β values normalized to a standard number of bacteria present in the host cell.

## Reverse transcription quantitative PCR

Bacterial whole cell lysates were generated as described previously (see Western Blot methodology), and used as input for RNA purification using the RNeasy Plus Mini Kit (Qiagen). WCL (50μl) was combined with 550 μl RLT buffer containing 1% β-mercaptoethanol (Sigma-Aldrich) before being passed through the included gDNA eliminator column and processed as described by the manufacturer. Whereas, BMDM monolayers were directly lysed in wells of a 24-well plate in 350 μl RLT buffer containing 1% β-mercaptoethanol (Sigma-Aldrich) before passed through a gDNA eliminator column and processed as per the manufacturer's recommendation. The resulting bacterial and mammalian RNA was assessed for quality and concentration by absorbance at 260nm using a Nanodrop 2000 Spectrophotometer (ThermoScientific). Five-hundred ng to 1 μg of RNA was used as input for DNaseI treatment following the manufacturer's recommendation (amplification grade; Invitrogen). After DNase I treatment, equal volumes of RNA were used as input for reverse transcription with

Superscript III Reverse Transcriptase (Invitrogen), again following the manufacturer's recommendations. The resulting cDNA was utilized as input for quantitative PCR analysis on a StepOnePlus Real Time PCR System (Applied Biosystems) using the PowerUp SYBR Green Master Mix (Thermo Fisher Scientific). Raw Ct values were quantified using the ΔΔCt method.

## Statistical analysis

For each experiment, samples were examined using a Shapiro-Wilk test to identify if data were normally distributed. Normally distributed data were then further analyzed using parametric statistical methods such as a one-way analysis of variance (ANOVA). Data that were non normally distributed were analyzed using non-parametric statistical methods such as a Kruskal Wallis test. Statistical significance was defined as p-values $\leq 0.05$ (*** $p<0.001$, ** $p<0.01$, * $p<0.05$, 'ns' not significant, $p>0.1$). All statistical analyses were performed using R version 4.2.2.

## Results

### Genetic conservation of SecA2 between *M. marinum* and *M. tuberculosis*

We sought to explore the conservation of SecA2 and SecA2-dependent pathogenesis (via activation of host cytoplasmic RNA sensors) between *M. tuberculosis* and *M. marinum*. To start, we compared the genomic organization and percent protein identity of genes present in the *secA2* locus for each species using Mycobrowser and NCBI. We observed that the organization of genes present in the locus of each of these two species is very similar, with only two genes present in *M. tuberculosis* that are absent in *M. marinum*. Additionally, the amino acid sequences these genes encode share a percent identity ranging from 78–96%. Notably, SecA2 shares an 88% amino acid identity between the two species (Fig 1 and S1 Fig). This high degree of similarity at the genetic level combined with previous reports of shared function [29, 30, 43–46], provided strong evidence to further explore the conservation of SecA2-dependent Rig-I/MAVS activation in host cells.

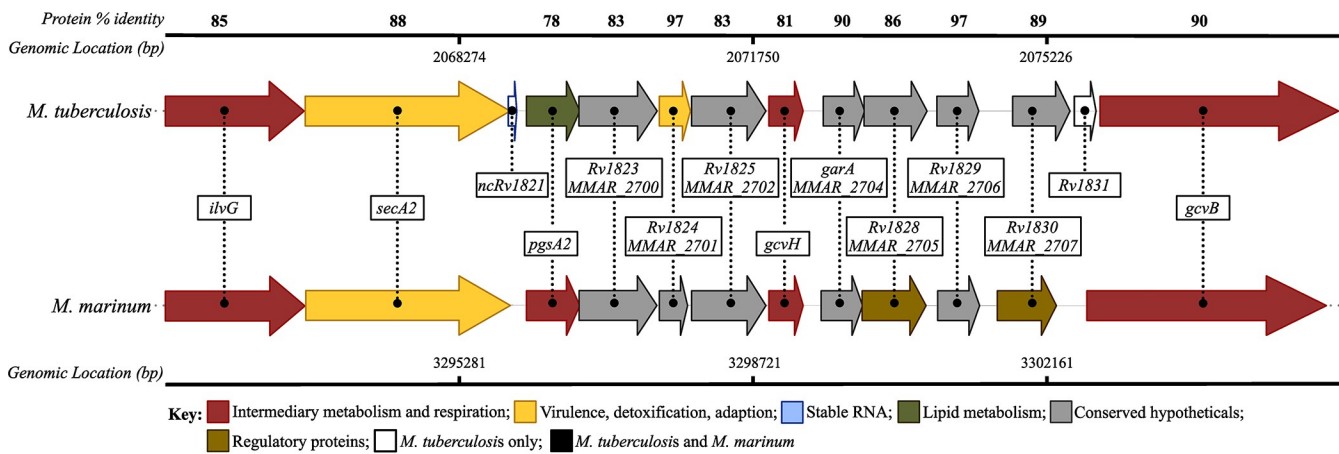

**Fig 1. Organization and conservation of *secA2* genomic locus in *M. marinum* and *M. tuberculosis*.** *M. tuberculosis* amino acid sequences (accession numbers from NC_000962.3) encoding proteins within the same locus as SecA2 were blasted against the *M. marinum* M genome (taxid: 216594) using NCBI Protein Blast. The percent protein identity for homologous proteins shared between the two mycobacterial species is depicted above each protein pair. Genomic locations, annotations, and gene function descriptions taken from NCBI and Mycobrowser. Fig design adapted from Chirakos et al., 2020 [37].

## Generation and validation of the Δ*secA2*, complementation and cross-complementation strains

To test our hypothesis that SecA2 dependent, Rig-I/MAVS activation and subsequent induction of IFN-β secretion by host macrophages is conserved in *M. marinum*, we generated an in-frame, unmarked deletion of *secA2*, removing amino acids 7 through 801. We also generated *M. marinum* complementation (Δ*secA2*/p*secA2*$_{MM}$) and *M. tuberculosis* cross-complementation strains (Δ*secA2*/p*secA2*$_{MT}$) to further explore this conservation and ensure observed phenotypes are the result of the deletion of *secA2* and not caused by polarity of the deletion on genes downstream. Generated strains were validated by PCR (Fig 2A–2D), qPCR (Fig 2E) and western blot analysis (Fig 2F).

PCR validation was conducted to check for (A) *secA2* at its endogenous location, (B) presence/absences of *secA2* in the genome, (C) *M. marinum* (*MMAR_2698*) and (D) *M. tuberculosis* (*Rv1821*) versions of the gene. Our results show the presence of *secA2* within the genome of every strain except Δ*secA2* (Fig 2A), as well as the presence of full length *secA2* in only the WT and Δ*esxBA* strains as well as the *M. marinum* gDNA positive control (top bands of 2758bp).

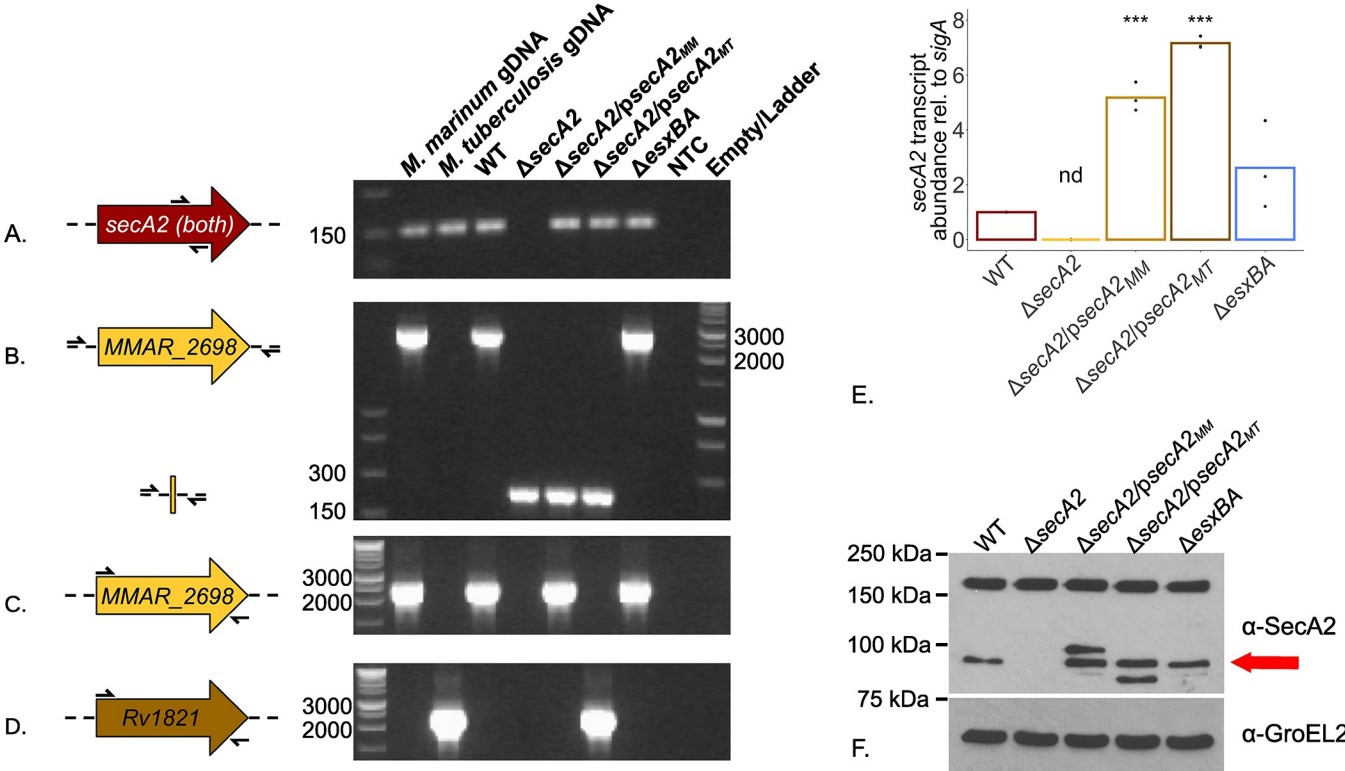

**Fig 2. Confirmation of *M. marinum* Δ*secA2* and complementation strains by PCR, qPCR, and western blot analysis.** *M. marinum* WCL was used as input for PCR confirmation of the presence of *secA2* within the genome (A; secA2_qPCR_F/R; 158 bp), the successful removal of *secA2* from it's endogenous location (B; secA2-X and secA2-Y primers; WT = 2758bp, Δ*secA2* = 370bp), *M. marinum secA2* (C; MMAR_2698 P1/P2; 2456 bp) and *M. tuberculosis secA2* (D; Rv1821_P1/P2; 2462bp). PCR products were separated on a 1.5% agarose gel and visualized with ethidium bromide on a BioRad GelDoc 1000. *M. marinum* and *M. tuberculosis* gDNA were used as positive controls, while nuclease free water served as a negative control where noted. Strains were further characterized by qPCR (E) and Western Blot (F). RNA was isolated from WCLs, reverse transcribed into cDNA, and used to assess the abundance of *secA2* relative to housekeeping gene *sigA* using the ΔΔct method. Statistical significance was determined using a one-way ANOVA followed by a Dunnett's Multiple Comparison Test relative to WT, with p-values ≤ 0.05 considered significant (*** p<0.001 and 'nd' not detected)(E). For western blot analysis, equal amounts of protein (60 μg) were added to lanes of an 8% SDS-PAGE-gel and separated by electrophoresis. Proteins were transferred onto a 0.45 μm, methanol-activated PVDF membrane and blotted for SecA2 and GroEL2 (loading control). The brightness and exposure for each whole blot was adjusted to 1.0 and 0.25 and is representative of three biological replicates. Red arrow indicates SecA2 to distinguish it from lower and higher molecular weight cross-reactive species (F).

Smaller products (370bp) present in the Δ*secA2*, Δ*secA2*/p*secA2*<sub>MM</sub>, and Δ*secA2*/p*secA2*<sub>MT</sub> indicate the deletion of the gene from it's endogenous location (Fig 2B). Deletion of *secA2* was further confirmed by Sanger sequencing (S2 Fig). PCRs with species specific *secA2* primers pairs confirmed the complementation and cross-complementation strain genotypes (Fig 2C–2D and S2 Table for sequences of all primers). Using the same primer pairs as Fig 2A, we additionally validated each strain by RT-qPCR. We observed no detectable *secA2* in the Δ*secA2* strain, while the Δ*secA2*/p*secA2*<sub>MM</sub> and Δ*secA2*/p*secA2*<sub>MT</sub> strains showed transcript levels of 5.18 (+/- 0.52 s.d.) and 7.16 (+/- 0.22 s.d.) relative to the housekeeping gene *sigA*, respectively (Fig 2E). The higher levels of *secA2* transcript abundance observed in the complement and cross-compliment strains are likely due to constitutive expression by the mycobacterial optimal promoter (MOP), rather than its native promoter. Finally, we also validated the strains by western blot analysis. Equal amounts of bacterial whole cell lysate protein was probed for SecA2 using a polyclonal α-SecA2 antibody [52] and GroEL2 (loading control). Our results indicate a roughly 90 kDa protein present in all strains except the Δ*secA2* (Fig 2F), indicated by the red arrow to differentiate it from lower and higher molecular weight species in the *M. marinum* and *M. tuberculosis* complementation strains. While we are unsure what these additional protein bands represent, they may be the result of post-translational modifications or degradation products of the SecA2 protein, as they are missing from the Δ*secA2* strain. Taken together, our results confirm the generation of an unmarked Δ*secA2* deletion as well as *M. marinum* and *M. tuberculosis* complementation and cross-complementation constructs.

## Impact of *secA2* on *in vitro* growth under specific media conditions

With our Δ*secA2*, Δ*secA2*/p*secA2*<sub>MM</sub>, and Δ*secA2*/p*secA2*<sub>MT</sub> strains built and validated, we sought to characterize the impact *secA2* has on growth *in vitro* and compare this to previously published data. We characterized growth *in vitro* under nutrient rich conditions (7H9 supplemented with 10% OADC and 0.2% tyloxapol) and fit logistic growth curves to the resulting data using the R package Growthcurver (S3 and S4 Figs). Our results illustrate a significant growth defect in the Δ*secA2* strain (p<0.001) that was either completely or partially restored by complementation with either the *M. marinum* or *M. tuberculosis* version of the gene (Fig 3, S3, S4A and S4B Figs). While the data represented in Fig 3 are based on growth curves completed with the same batch of OADC, we observed that the degree of complementation was variable across batches of OADC (S5 Fig). Additionally, when grown in minimal media (7H9 + 0.5% glycerol + 0.5% glucose + 0.1% tween 80) a severe aggregation phenotype was observed in the Δ*secA2* strain (S6 Fig). These data are contrary to previous observations in Δ*secA2* *M. tuberculosis* [28], *M. marinum* [29] and *M. smegmatis* [53] where *in vitro* growth curves in nutrient rich liquid media showed no growth defect. However, slower growth and changes in morphology have been described in these studies under specific media conditions [28, 29, 53].

Two such observations of *M. marinum* Δ*secA2*-related growth deficiencies involved increased susceptibility to antibiotics specifically targeting the cell wall [43] and sensitivity to SDS [29], both of which suggest a cell wall synthesis defect. To examine if this phenotype was present in our Δ*secA2* strain, we tested it for sensitivity to 0.05–0.2% SDS. In support of previous work by Watkins et al., our Δ*secA2* strain also showed an observable growth defect in SDS (0.052 relative growth +/- 0.037 s.d.; Fig 3B) as compared to growth in the same media lacking SDS. This phenotype was partially or fully rescued to WT levels (1.61 relative growth +/- 1.17 s.d.) by complementation with the *M. marinum* or *M. tuberculosis* version of *secA2* (1.88 relative growth +/- 0.98 s.d.; Fig 3B and 3C and S7A and S7B Fig). Similar to growth in nutrient rich media, the degree of complementation varied across different media preparations.

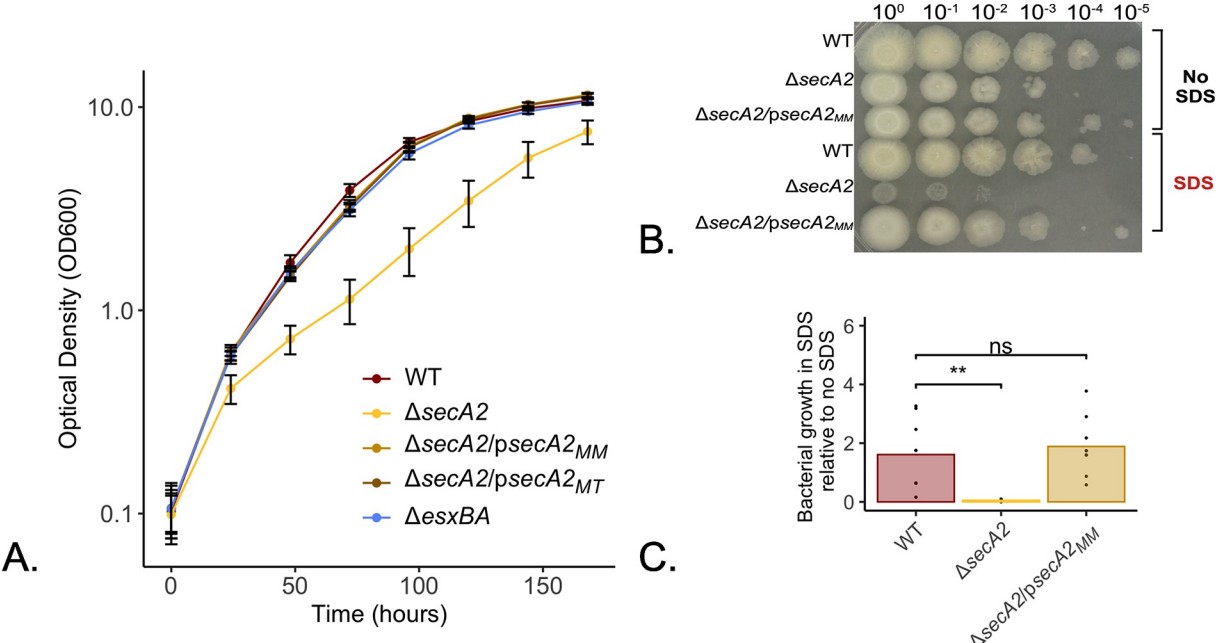

**Fig 3.** *M. marinum ΔsecA2* **strain has attenuated growth** *in vitro* **and sensitivity to SDS.** *M. marinum* strains were grown to exponential phase in 7H9 supplemented with 10% OADC and 0.2% tyloxapol before being subcultured into fresh media at an optical density ($OD_{600}$) of 0.05. Bacterial growth was monitored by measuring the $OD_{600}$ every 24-hours for 7 days and plotted on a log transformed scale (A). The same strains were examined for growth in sodium dodecyl-sulfate (SDS) by subculturing exponential phase bacteria into 7H9 supplemented with 10% OADC and 0.2% tyloxapol with and without 0.2% SDS at an OD600 of 0.8. Twenty-four hours later, these cultures were serially diluted and plated onto 7H11 agar plates supplemented with 10% OADC in technical triplicate. Images are representative of three biological replicates each plated in technical triplicate (B). Colony forming units were quantified from agars plates following incubation at 32°C with 5% $CO_2$ for one week (C). Statistical significance was determined using a one-way ANOVA followed by a Dunnett's test for multiple comparison relative to WT, with p-values $\leq 0.05$ considered significant (C only).

## SecA2 is not required for the intracellular growth of *M. marinum* in BMDMs

Upon phagocytosis by alveolar macrophages, pathogenic mycobacteria are placed into a phagosome. To evade degradation by the maturation and acidification of these membranous compartments, mycobacteria have evolved mechanisms to subvert this process. Protein secretion systems including ESX-1 and SecA2 play vital roles in the bacterium's ability to survive intracellularly. For example, the ESX-1 protein secretion system is responsible for the translocation of substrate(s) that act to permeabilize the phagosomal membrane, providing the bacterium access to the host cytosol, and with it, new opportunities for nutrient acquisition and host manipulation [42, 54–56]. The SecA2-dependent Rig-I/MAVS activation of INF-β signaling and secretion in *M. tuberculosis* has been shown to be dependent on a functional ESX-1 protein secretion system [11]. Without ESX-1, the bacterially derived RNAs which activate Rig-I/MAVS, are trapped within the phagosomal compartment. For this reason, an ESX-1 deletion strain (*ΔesxBA*) is included as a control for attenuated intracellular growth and low IFN-β inducing bacteria in all infection based studies.

Removal of *secA2* has been shown to attenuate bacterial survival in mouse [46] and zebrafish [29, 43] models infected with *M. tuberculosis* or *M. marinum* M or E11, respectively. Intracellular models using primary, mouse bone marrow derived macrophages, reciprocate this data in *M. tuberculosis* [46] but not for *M. marinum* [29]. We tested our *ΔsecA2 M. marinum* strain for intracellular growth in non-activated BMDMs over 96 hours (Fig 4). By fitting linear

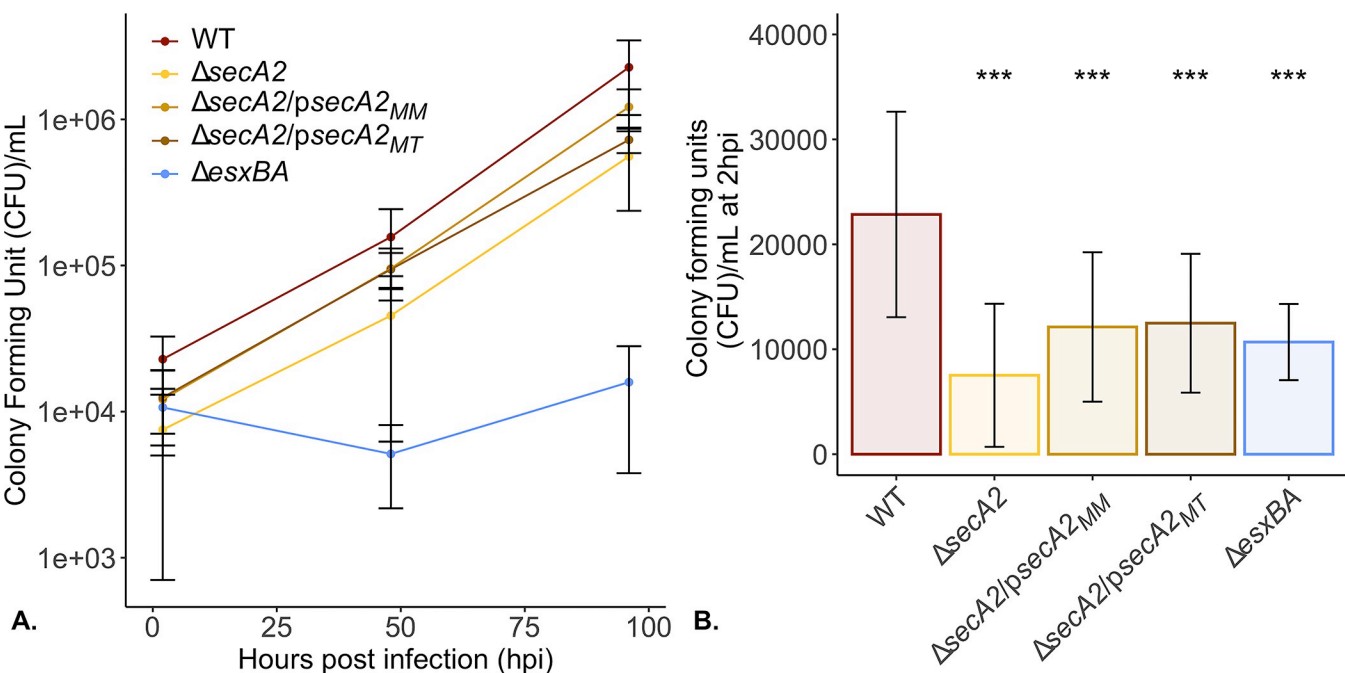

**Fig 4. Δ*secA2* uptake relative to bacterial input and growth in BMDMs is comparable to WT.** *M. marinum* strains were grown to exponential phase in 7H9 supplemented with 10% OADC and 0.2% tyloxapol and used to infect triplicate wells of BMDMs, at an MOI of 0.2, for 2 hours. At 2, 48, and 96 hours post infection (hpi), macrophage monolayers were lysed and plated onto 7H11 agar supplemented with 10% OADC in technical triplicate. One week after plating, colony forming units were enumerated for each timepoint (A). Bacterial numbers present at 2hpi were plotted separately (B) to better visualize differences in bacterial counts. Uninfected wells were included as a control for across well contamination (not shown, no contamination observed) and the Δ*esxBA* strain as a control for attenuated growth. Statistical significance was determined using a Kruskal-Wallis test followed by a Wilcoxon Rank Sum test for pairwise comparison relative to WT, with p-values ≤ 0.05 considered significant (*** p<0.001).

regression models to colony forming unit counts for each time point, we were able to compare intracellular growth rates across bacteria strains (S8, S9A and S9B Fig). In accordance with data presented by Watkins et al., we observed no attenuation of growth in the Δ*secA2* as compared to the wild type and complemented strains. However, we did observe a 10 fold reduction (0.0042 +/- 0.006 s.d.) in growth rate for the Δ*esxBA* strain as compared to the WT (0.049 +/- 0.004 s.d.).

From this data, we also observed strain-specific differences in the number of bacteria present in host macrophages 2hpi (Fig 4A). When infecting macrophage monolayers, we utilized an established, OD-based calculation that assumes at an $OD_{600}$ = 1, there are approximately $7.7 \times 10^7$ bacteria present in one mL of bacterial suspension [57]. Applying this calculation across our five strains, it was notable that the number of colony forming units present in host macrophages infected with Δ*secA2 M. marinum* at 2hpi was on average 3 fold lower than those infected with WT bacteria (Fig 4B). When compared to the number of bacteria added to macrophages (bacterial input), our data indicate that the observed reduction in number of bacteria taken up is not likely due to a defect in the macrophage's ability to phagocytose the bacteria (S10 Fig) but due to an underestimation of the number of bacteria present in suspension at a given optical density.

### IFN-β secretion from WT BMDMs is not SecA2-dependent in *M. marinum*

The results observed in Fig 4 suggested that it is important to take into account the number of intracellular bacteria present in macrophages following infection when comparing levels of

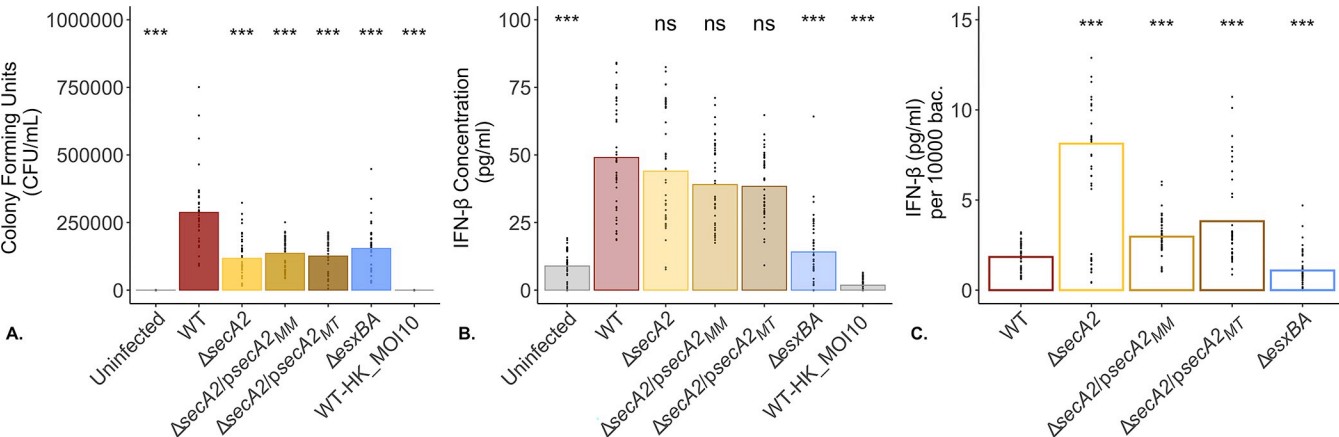

**Fig 5. Δ*secA2 M. marinum* induces high levels of IFN-β from WT BMDMs.** To evaluate the IFN-β response from wild type BMDMs when levels of intracellular bacteria are equal, log-phase *M. marinum* cultures were used to infect wild type BMDMs at an MOI of 1. Following a 2 hour infection, macrophage monolayers were washed three times with PBS before being lysed and plated onto 7H11 + 10% agar plates in technical triplicate (A) or returned to the incubator in fresh media until twenty-four hours post infection. At twenty-four hours, macrophage culture supernatants were removed and examined for secreted levels of IFN-β by ELISA (B). To account for variability in the number of bacteria present at 2hpi, IFN-β levels were normalized to a standard number of bacteria (C). IFN-β Heat killed WT *M. marinum* was added to macrophages at an MOI of 10 as a control for bacterial cell lysis. Data is representative of 10 biological replicates. Statistical significance was calculated using the non-parametric Kruskal-Wallis test followed by pairwise comparison with a Wilcoxon Rank Sum test relative to WT. Statistical significance was defined as p-values ≤ 0.05; ***p<0.001, **p<0.01, *p<0.05, ns = not significant (p>0.05).

secreted IFN-β. Therefore, when examining BMDMs for secreted levels of IFN-β, we infected these cells using the same OD-based calculation as before, and additionally plated macrophage whole cell lysates at the end of the infection period (2hpi). This enabled us to quantify the number of intracellular bacteria following a 2 hour infection (Fig 5A) and secreted levels of IFN-β (Fig 5B) separately. Crucially, it also allowed us to correct for differences in numbers of intracellular bacteria to examine secreted levels of IFN-β based on equal numbers of bacterial colony forming units (Fig 5C).

Using this adapted methodology we observed an average 2.2 fold reduction in bacterial CFUs for all other bacterial strains as compared to WT (p<0.001; Fig 5A). Despite having fewer bacteria present at 2hpi, the Δ*secA2*, Δ*secA2*/p*secA2*$_{MM}$, Δ*secA2*/p*secA2*$_{MT}$ induced IFN-β at levels comparable to WT (Fig 5B), while the Δ*esxBA* strain induced significantly less IFN-β (14.13 pg/mL +/- 12.08 s.d. versus 49.07 pg/mL +/- 20.29 s.d.) (Fig 5B). When normalized to an equal number of bacterial CFUs (10,000) at 2hpi, our results showed that the Δ*secA2* and both complementation strains induced higher levels of IFN-β from host macrophages as compared to WT *M. marinum* (Fig 5C). It additionally confirmed previous findings showing the Δ*esxBA* as a low IFN-β inducing control following normalization. As the removal of *secA2* impacts cell wall integrity, we included heat killed wild type *M. marinum* to control for IFN-β secretion induced by lysed mycobacterial components and saw levels of secreted IFN-β (1.71 pg/mL +/- 2.14 s.d.) comparable to uninfected controls (8.93 +/- 5.57 s.d. pg/mL). We additionally assessed the impact increasing numbers of Δ*secA2 M. marinum* (with MOIs ranging from 1 to 10) had on secreted levels of IFN-β (S11A–S11C Fig). Higher CFU counts at 2hpi induced a strong linear increase in the IFN-β in response for each biological replicate, although with variation across replicates in terms of the magnitude of the response elicited (slope of the line; S12 Fig). Notably, when normalized, Δ*secA2 M. marinum* induced an average amount of IFN-β around 7 times higher than WT for all MOIs tested (S11C Fig).

As the kinetics of an *M. marinum* infection are faster than that of *M. tuberculosis*, with a faster doubling rate and quicker access to the host cytosol, we also examined the host response to *M. marinum* infection earlier in the infection by quantitative PCR (S13 Fig). We assessed

BMDMs infected with out panel of *M. marinum* strains for relative levels of transcript abundance for IFN-β, the transcription factor Irf7, and upstream signalling molecule Rig-I relative to the housekeeping gene GAPDH at 8 hours post infection. Results from this assay showed no difference in transcript abundance for any of the genes tested across all strains except Δ*esxBA*, where there was a significant reduction in transcript abundance for all three genes as compared to WT (S13 Fig).

## IFN-β secretion from MAVS^-/- BMDMs is not SecA2-dependent in *M. marinum*

Following activation of Rig-I by immunostimulatory RNAs (such as those derived from bacteria and viruses), Rig-I undergoes a conformational change that enables its interaction with MAVS. MAVS is an essential adapter protein that relays this activation signal onwards, ultimately inducing the transcription of high levels of type I interferons [27, 58]. While this response was originally attributed to defense against viruses, it is increasingly recognized for its complex role in bacterial infections [1, 6, 59–61]. In *M. tuberculosis*, for example, it has been shown that MAVS^-/- deficient mice are better able to control bacterial proliferation in the lungs six weeks after aerosol challenge with wild type *M. tuberculosis* [11].

We therefore sought to examine the contribution of this signaling pathway to the INF-β response in *M. marinum* and explore how this IFN-β response is impacted by the presence or absence of a functional SecA2-protein secretion system using the same experimental model utilized in Fig 5 but with MAVS^-/- BMDMs. Similar to our observations with WT BMDM, we observed an approximately 2.5 fold reduction in bacterial CFUs 2 hpi across all bacterial strains compared to WT (Fig 6A). Despite fewer bacteria measured at 2hpi in the Δ*secA2*, Δ*secA2*/p*secA2*$_{MM}$, and Δ*secA2*/p*secA2*$_{MT}$ strains, we observed levels of secreted IFN-β comparable to WT (Fig 6B). While the Δ*esxBA* strain had CFU numbers comparable to Δ*secA2* and its complementation strains, we measured significantly less secreted IFN-β by ELISA (15.1 pg/mL +/- 6.7 versus 56.2 pg/mL +/- 14.4 for WT and 51.2 pg/mL +/- 28.1 for Δ*secA2*).

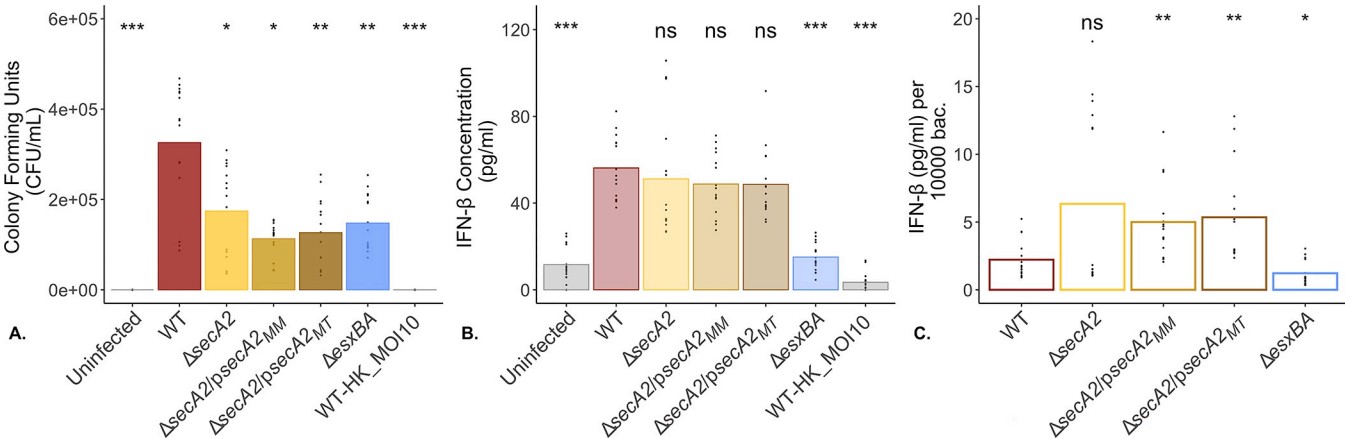

**Fig 6. Δ*secA2 M. marinum* induces high levels of IFN-β from MAVS^-/- BMDMs.** To evaluate the contribution of the MAVS intracellular RNA signaling pathway in the IFN-β response and examine whether this response is dependent on a functional SecA2 protein secretion system, log-phase *M. marinum* cultures were used to infect MAVS^-/- BMDMs at an MOI of 1. Following a 2 hour infection, macrophage monolayers were washed three times with PBS before being lysed and plated onto 7H11 + 10% agar plates in technical triplicate (A) or returned to the incubator in fresh media until twenty-four hours post infection. At twenty-four hours, macrophage culture supernatants were removed and examined for secreted levels of IFN-β by ELISA (B). To account for variability in the number of bacteria present at 2hpi, IFN-β levels were normalized to a standard number of bacteria (C). Heat killed WT *M. marinum* was added to macrophages at an MOI of 10 as a control for bacterial cell lysis. Data is representative of 5 biological replicates. Statistical significance was calculated using the non-parametric Kruskal-Wallis test followed by pairwise comparison with a Wilcoxon Rank Sum test relative to WT. Statistical significance was defined as p-values ≤ 0.05; ***p<0.001, **p<0.01, *p<0.05, ns = not significant (p>0.05).

When normalized to 10,000 bacterial CFUs at 2hpi, the Δ*secA2*, complementation and cross-complementation strains were shown to induce INF-β secretion from host MAVS[-/-] macrophages to concentrations equal to or greater than levels than WT (Fig 6C). Consistent with previous observations, the Δ*esxBA* strain induced significantly lower levels of secreted IFN-β from macrophages infected with WT *M. marinum* (p<0.01)(10,20). To ensure the results reported above were representative of a MAVS[-/-] BMDM model, we additionally probed macrophage WCLs, derived from the same infection experiments, for MAVS by western blot (S14 Fig). MAVS was present in WT but not in the MAVS[-/-] BMDMs, confirming the genotype of each cell type used.

## Discussion

In this study we sought to examine the conservation of a SecA2-dependent pathogenesis mechanism utilized by *M. tuberculosis* in a related, BSL2 mycobacterium. Following the construction of a Δ*secA2 M. marinum* knockout and complementation strains, we first examined each bacterial strain's growth *in vitro* and within host cells to compare them to previous literature. Here, we observed an *in vitro* growth defect in the Δ*secA2* strain when cultured in both nutrient rich liquid media and in the presence of SDS. Intracellularly, we observed that bacterial growth was not impacted by the loss of a functional SecA2-protein secretion system in unactivated, WT BMDMs. This intracellular growth experiment additionally revealed an underestimation in the number of Δ*secA2*, its complementation strains, as well as the Δ*esxBA M. marinum* strain added to macrophage monolayers compared to WT *M. marinum*. Moving forward, we included an additional plating step when examining the type I interferon response in BMDMs following infection with our bacterial strains to compensate for variations in numbers of intracellular bacteria present at 2hpi. This enabled the normalization of IFN-β concentration to a standard number of bacteria. Using an unactivated murine, bone-marrow derived macrophage infection model we observed that IFN-β was induced and secreted from macrophages infected with Δ*secA2 M. marinum* at levels equal to or higher than WT bacteria. Additionally, we found no evidence that the Rig-I/MAVS cytosolic sensing pathway contributed to this IFN-β response in a SecA2-dependent manner.

Some of these observations were consistent with previously reported data, including sensitivity to SDS by the Δ*secA2* strain [29]. However, our *in vitro* growth curves are contrary to previously published research which show no growth defect in *M. smegmatis* [53], *M. tuberculosis* [28], or *M. marinum* [29] under similar growth conditions. Notably, growth defects and morphological changes have been observed in *M. tuberculosis* [28] and *M. smegmatis* [53] Δ*secA2* strains on nutrient rich Muller-Hinton agar plates. It remains unclear what is responsible for the differences in the Δ*secA2* growth curves between studies but we hypothesize that small variations in growth media can have a significant effect on Δ*secA2* strain survival/replication. This is supported by our data where we observed significant differences in the growth curves of the Δ*secA2* strain under different media conditions. It is likewise possible that there are in fact species specific functions of SecA2 that impact bacterial growth *in vitro*. Proteomic analysis conducted by Feltcher et al. in 2015, revealed an enrichment for two *M. tuberculosis* Mce family transporters in the absence of SecA2 [44]. These transporters are predicted to be involved in fatty acid and cholesterol import [62, 63], and suggest that SecA2 dependent changes in cell wall composition may impact bacterial growth and metabolism. Additionally, genes involved in intermediary metabolism and respiration are well represented in the *secA2* locus, further suggesting that removal of this gene could contribute to changes in metabolism, nutrient acquisition, and subsequent growth, which may explain the high degree of variability sometimes observed in this strain. However, similar experiments in *M. marinum* would be needed to define the impacts of the *secA2* locus on cell wall composition.

The *in vitro* growth defect we observed in the Δ*secA2* strain under nutrient rich conditions did not impact this strain's growth within host macrophages as compared to WT. This data is in support of similar findings by Watkins et al., where they infected both IFN-γ activated and unactivated WT BMDMs with WT, Δ*secA2*, and a Δ*secA2* complemented *M. marinum* M strain and measured similar growth rates over 96 hours for all stains under both conditions tested [29]. While these results are consistent, they differ from what has been observed for *M. tuberculosis*. Previous research using a Δ*secA2 M. tuberculosis* strain demonstrated that *secA2* contributes to attenuated bacterial growth within murine bone marrow derived macrophages over time [46], and implies that, unlike in *M. tuberculosis*, *secA2* does not contribute to *M. marinum* survival in murine bone-marrow derived macrophages.

Another observation we made through the intracellular survival experiment was an under-estimation in the number of Δ*secA2*, complementation and Δ*esxBA* strains as compared to WT present inside host cells at 2hpi. Therefore, we took this differential uptake into consideration in our subsequent studies examining the IFN-β response using our generated bacterial strains. When examined for their ability to induce IFN-β in WT and MAVS$^{-/-}$ BMDMs, we again observed fewer bacteria 2hpi as compared to WT for the Δ*secA2*, complemented and Δ*esxBA* strains. Additionally, we observed secreted levels of IFN-β equal to or higher than WT bacteria following infection with the Δ*secA2* strain when normalized to 10,000 bacterial CFUs. These findings are not consistent with work in *M. tuberculosis* [11] or *Listeria monocytogenes* (a Gram-positive bacteria with a SecA2-protein secretion system) [61], where bacterially derived RNAs gain access to and activate host Rig-I/MAVS cytosolic sensors in a SecA2-mediated, ESX-1 (*M. tuberculosis*) or listeriolysin O (*L. monocytogenes*) dependent process.

As mentioned previously, an *M. marinum* M Δ*secA2* strain has been shown to have attenuated survival in zebrafish and a small but significant attenuation for growth in mouse tails 21 days post infection, but not in murine activated or unactivated BMDMs [29]. This same study also observed, at 3 weeks post infection in mice and 11 days post infection in zebrafish, there was significantly less secreted TNF-α and TNF-α transcript abundance, respectively, following infection with Δ*secA2 M. marinum* as compared to WT. However, they additionally reported that secreted levels of TNF-α were unaffected by the loss of a functional SecA2 protein secretion system in BMDMs infected with the same strains. In studies examining bacterial growth and TNF-α cytokine response in *M. tuberculosis*, attenuated bacterial growth *in vivo* correlated with attenuated growth and reduced levels of secreted TNF-α from isolated, unactivated BMDMs infected with Δ*secA2* as compared to WT strains [64]. A similar observation is emerging for the SecA2-dependent induction of IFN-β, where bacterial growth and levels of secreted cytokine are significantly reduced in BMDMs infected with *M. tuberculosis* CDC1551 Δ*secA2* but not *M. marinum* M Δ*secA2* (this study, 58). A possible explanation for this uncoupling between *in vivo* and *ex vivo* observations in *M. marinum* Δ*secA2* strains but not *M. tuberculosis* include species specific differences in genome size, optimal growth requirements, doubling time, motility, and natural host niches [54, 65, 66]. These differences have the potential to contribute to variations in host:pathogen interactions and virulence mechanisms [35, 67]. While it remains possible that SecA2-dependent release of bacterially derived RNAs is not conserved in *M. marinum* under any condition, it may instead reflect a limitation in the model organism utilized in this study.

Future work using the strains generated for this paper might include proteomic analysis to explore changes in cell wall and cytoplasmic protein abundance, which may provide insight into factors contributing to observed changes in *in vitro* growth. It may also be worth testing the conservation of SecA2-dependent release of bacterial RNAs and activation of host cytosolic sensors in a different host organism. Here, zebrafish may be a more realistic model organism that could provide a stronger conclusion on the conservation of SecA2-dependent

pathogenesis not only because they are a natural host for *M. marinum* but also because they have been shown to possess a strong type-I interferon system. Similar to murine BMDMs, this interferon system is also driven by Rig-I and MAVS orthologues, which provide protection against RNA and DNA virus replication [68].

Taken together, our results highlight the importance of measuring bacterial uptake alongside experiments aimed at examining changes in the host cell secretome, especially when including strains with known growth defects. They additionally indicate that, under the conditions tested, the SecA2-dependent release of bacterially derived RNAs and subsequent induction of a type I interferon response that has been shown to be an important pathogenesis mechanism in *M. tuberculosis* is not conserved in *M. marinum*.

## Supporting information

**S1 Table. A list of bacterial strains used in this study.**
(PDF)

**S2 Table. A list of oligonucleotide primers used throughout this paper.** All primers were purchased through Invitrogen.
(PDF)

**S3 Table. A list of plasmids used in this study.**
(PDF)

**S4 Table. Primary and secondary antibodies used for western blot analysis in this study.**
(PDF)

**S1 Fig. *M. marium* and *M. tuberculosis* SecA2 shares 88.34% sequence identity.** Sequence alignment of *M. marinum* M (accession no.: ACC41141.1) and *M. tuberculosis* (QNF05793.1) SecA2. Sequences were obtained from NCBI and aligned using Clustal Omega 1.2.4. Conservation in amino acid sequence is denoted by an asterix (*) while changes resulting in strongly, weakly, or no similar chemical properties are denoted by a colon (:), period (.), or left blank.
(PDF)

**S2 Fig. Confirmation of the ΔsecA2 *M. marinum* strain by Sanger sequencing.** Primer pairs X & Y were used to amplify *secA2* from its endogenous location in the genome in the ΔsecA2 *M. marinum* strain. Purified PCR products were sequenced by the Genomics Core at the University of Notre Dame using the secA2-X primer. Deletion of *secA2* was confirmed using the chromatogram viewer FinchTV version 1.5.0. Arrows above the chromatogram indicate the primer sequences (secA2-B and secA2-C) and encoded AflII restriction site used in the generation of the knockout.
(PDF)

**S3 Fig. Δ*secA2 M. marinum* has a lower growth rate in nutrient rich media than WT and complemented strains.** Representative images from three independent biological replicate growth curves. A logistic growth curve model was fit to average bacterial growth optical density (OD600) values (from three technical replicates each measured in technical triplicate). Individual plot titles represent the bacterial strain, biological replicate number, and growth rate (r). Best fit line is indicated in red.
(PDF)

**S4 Fig. Bacterial *in vitro* growth measurements were fit to a logarithmic growth curve using the Growthcurver R package.** The resulting parameter outputs (A; representative of the average for each biological replicate) were used to compare bacterial growth rates (r). These

growth rates were plotted (B), with points representative of growth rates for each technical replicate (n = 3) for each biological replicate (n = 3) for a total of 9 data points per stain. Bar heights represent the average value of all 9 data points. Statistical significance was assessed using a one-way ANOVA followed by a Dunnett's pairwise comparison relative to WT. *** p<0.001.
(PDF)

**S5 Fig. Variation is Δ*secA2* growth in different batches of OADC.** *M. marinum* strains were grown to exponential phase in 7H9 supplemented with 10% OADC and 0.2% tyloxapol before being subcultured into fresh media at an optical density ($OD_{600}$) of 0.05. Bacterial growth was monitored by measuring the $OD_{600}$ every 24-hours for 7 days.
(PDF)

**S6 Fig. Δ*secA2* bacteria aggregate in the presence of minimal media.** *M. marinum* strains were grown to exponential phase in 7H9 supplemented with 0.1% tween80 before being subcultured into fresh media at an optical density ($OD_{600}$) of 0.05. Upon subculturing, a pronounced aggregation phenotype was observed. Δ8 is shorthand for Δ*secA2*.
(PDF)

**S7 Fig. *M. marinum* Δ*secA2* strain is sensitivity to SDS.** *M. marinum* strains were grown to exponential phase in 7H9 supplemented with 10% OADC and 0.2% tyloxapol before being subcultured into fresh media with and without 0.05% SDS at an OD600 of 0.8. Twenty-four hours later, these cultures were serially diluted and plated onto 7H11 agar plates supplemented with 10% OADC in technical triplicate. Image is representative of two biological replicates each plated in technical triplicate (A). Colony forming units were quantified from agars plates following incubation at 32˚C with 5% $CO_2$ for one week and bacterial growth in SDS was quantified relative to growth in nutrient rich media lacking SDS. Inset represents the same data with an adjusted y-axis (B). Statistical significance was calculated using the non-parametric Kruskal-Wallis test followed by pairwise comparison with a Wilcoxon Rank Sum test relative to *M. marinum* Δ*secA2*. *Δ8 = ΔsecA2;* *** p-values ≤ 0.001.
(PDF)

**S8 Fig. Triplicate wells of WT BMDMs were infected with *M. marinum* strains at an MOI of 0.2.** At 2, 48, and 96-hours post infection, macrophage monolayers were washed three times and lysed in sterile water containing 0.2% tyloxapol. Whole cell lysate dilutions were plated in technical triplicate and grown at 32˚C for 1 week. Colony forming units were counted and linear equations were fit to log transformed colony counts (red line) for each of the three biological replicates. Individual plot titles represent the bacterial strain, biological replicate number, and best fit line equation for average data from each biological replicate.
(PDF)

**S9 Fig. Bacterial intracellular growth colony counts were fit to a linear regression model.** The resulting parameter outputs (A; representative of the average for each biological replicate) were used to compare bacterial growth rates (r). These growth rates were plotted (B), with points representative of growth rates for each biological replicate (n = 3). Bar heights represent the average value of all 9 data points. Statistical significance was assessed using a one-way ANOVA followed by a Dunnett's pairwise comparison relative to WT. *** p<0.001.
(PDF)

**S10 Fig. The number of bacterial colony forming units present at 2hpi (uptake) relative to the number of bacteria added to each well at the start of the infection (input).** Statistical significance was assessed using a one-way ANOVA followed by a Dunnett's pairwise comparison

relative to WT. * p<0.05.
(PDF)

**S11 Fig. WT BMDMs secrete increasing concentrations of IFN-β relative to increased MOIs.** To evaluate the IFN-β response from wild type BMDMs when levels of intracellular bacteria are added in increasing amounts, log-phase *M. marinum* cultures were used to infect wild type BMDMs at various MOIs. Following a 2 hour infection, macrophage monolayers were washed three times with PBS before being lysed and plated onto 7H11 + 10% agar plates in technical triplicate (A) or returned to the incubator in fresh media until twenty-four hours post infection. At twenty-four hours, macrophage culture supernatants were removed and examined for secreted levels of IFN-β by ELISA (B). To account for variability in the number of bacteria present at 2hpi, IFN-β levels were normalized to a standard number of bacteria (C). IFN-β Heat killed WT *M. marinum* was added to macrophages at an MOI of 10 as a control for bacterial cell lysis. Data is representative of 10 biological replicates. Statistical significance was calculated using the non-parametric Kruskal-Wallis test followed by pairwise comparison with a Wilcoxon Rank Sum test relative to WT. Statistical significance was defined as p-values $\leq 0.05$; ***p<0.001, **p<0.01, *p<0.05, ns = not significant (p>0.05). All infections were conducted at and MOI of 1 unless otherwise noted.
(PDF)

**S12 Fig. Secreted levels of IFN-β increase with increasing Δ*secA2* *M. marinum* MOI.** BMDMs were infected with increasing amounts of Δ*secA2* *M. marinum* (MOIs of 1, 2.5, 5, and 10). Average levels of secreted IFN-β present in the supernatant 24 hours after the start of the infection were plotted against average CFUs plated at the end of the infection period (2hpi). A linear model was applied to each replicate to determine a best fit line (red). Equations defining the best fit line and associated $R^2$ value are provided above each graph.
(PDF)

**S13 Fig. Δ*secA2* *M. marinum* induces similar levels of IFN-β, Irf7, and Rig-I as WT and complemented strains.** RNA was isolated from BMDMs 8hpi with indicated *M. marinum* strains, purified, reverse transcribed, and examined for relative abundance of transcripts for IFN-β (left panel), Irf7 (middle panel), and Rig-I (right panel) relative to the housekeeping gene GAPDH for two biological replicates examined in technical duplicate. **p<0.01; all statistics were assessed relative to WT levels.
(PDF)

**S14 Fig. Confirmation of MAVS knockout in MAVS[-/-] BMDMs.** Upon the removal of culture supernatants for ELISA analysis (Figs 5 and 6), macrophage monolayers from the same infections were lysed in 75ul RIPA lysis buffer. WCLs were quantified by BCA and 15 μg of protein were separated on an 8% SDS-PAGE gel. Proteins were transferred to a 0.45 μM, methanol-activated PVDF membrane and probed for Rig-I, MAVS, or β-actin (loading control).
(PDF)

**S1 Raw images.**
(PDF)

# Acknowledgments

Thank you to members of the J.S. Schorey lab for their support, especially William McManus for critical reading of the manuscript, and to members of the P.A. Champion lab for feedback on experimental results and expertise with *M. marinum*. We are grateful to Sean M. Cavany

for computational assistance and critical reading of the manuscript, and Miriam Braunstein for the α-SecA2 antibody.

## Author Contributions

**Conceptualization:** Lindsay G. Serene, Patricia A. Champion, Jeffrey S. Schorey.

**Data curation:** Lindsay G. Serene.

**Formal analysis:** Lindsay G. Serene.

**Funding acquisition:** Lindsay G. Serene, Patricia A. Champion, Jeffrey S. Schorey.

**Investigation:** Lindsay G. Serene.

**Methodology:** Lindsay G. Serene, Kylie Webber, Patricia A. Champion, Jeffrey S. Schorey.

**Project administration:** Lindsay G. Serene.

**Resources:** Patricia A. Champion, Jeffrey S. Schorey.

**Software:** Lindsay G. Serene.

**Supervision:** Patricia A. Champion, Jeffrey S. Schorey.

**Visualization:** Lindsay G. Serene, Jeffrey S. Schorey.

**Writing – original draft:** Lindsay G. Serene.

**Writing – review & editing:** Lindsay G. Serene, Kylie Webber, Patricia A. Champion, Jeffrey S. Schorey.

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
