## [Decision Letter · Decision Letter 0]

24 Apr 2023

PONE-D-23-02235Mycobacterium tuberculosis SecA2-dependent activation of host Rig-I/MAVs signaling is not conserved in Mycobacterium marinumPLOS ONE

Dear Dr. Serene,

Thank you for submitting your manuscript to PLOS ONE. After careful consideration, we feel that it has merit but does not fully meet PLOS ONE’s publication criteria as it currently stands. Therefore, we invite you to submit a revised version of the manuscript that addresses the points raised during the review process.

We look forward to receiving your revised manuscript.

Kind regards,

Atul Vashist, PhD

Academic Editor

PLOS ONE

“This work was supported by the Eck Institute for Global Health Graduate Student Fellowship, awarded to LGS. PAC is supported by the National Institutes of Health under award numbers AI156229, AI106872, AI149147, and AI149235. The content of this article is solely our responsibility and does not necessarily represent the official views of the National Institutes of Health.”

Reviewers' comments:

Reviewer's Responses to Questions

**Comments to the Author**

1. Is the manuscript technically sound, and do the data support the conclusions?

Reviewer #1: Partly

Reviewer #2: Yes

2. Has the statistical analysis been performed appropriately and rigorously? 

Reviewer #1: Yes

Reviewer #2: Yes

3. Have the authors made all data underlying the findings in their manuscript fully available?

Reviewer #1: Yes

Reviewer #2: Yes

4. Is the manuscript presented in an intelligible fashion and written in standard English?

Reviewer #1: Yes

Reviewer #2: Yes

5. Review Comments to the Author

Reviewer #1: The authors aim to show that the SecA2 secretion system of mycobacterium marinum does contribute to IFN-beta production. This would make for a more convenient model on BSL-2 to study this process in mycobacteria. Unfortunately no SecA2 depended type I interferon production was observed in M. marinum. Even replacement of the SecA2 secretion system of marinum with the mycobacterium tuberculosis version, does not result in SecA2 dependent IFN-beta production. Furthermore the authors correlate the amount of IFN-beta produce to the CFU at time 2hous post infection to relate IFN-beta production to the number of bacteria to circumvent any bias. This could have some additional value, since uptake and e.g. clumping of the bacteria can all affect the CFU recovered at 2h post infection and therefore the response measured. The validity of the normalization is however uncertain. The authors present negative data, that could be interesting, but should provide a more evidence to support their claims.

Major Concerns:

1) The functionality of the complementation strains is tested to a limited extend, especially for the complementation with the SecA2 from tuberculosis. In figure 3B complementation of Sec2A from tuberculosis is missing and should be tested. As an alternative the secretion of e.g. PknG could be tested.

2) Time of measurement at 24h. M. marinum is a more fast paced pathogen. It grows faster, but also escapes (has access to) earlier to the cytosol. In reference 11 (as mentioned in manuscript) in figure 3 it is clear that the IFN-beta response diminishes over time. Measuring at 24 hours might be too late to observe any difference in response. It is therefore imperative to measure earlier, e.g. 6-8 hours post infection. This can be done by either ELISA but also qPCR.

3) Normalization strategy. All strains except for the EsxBA mutant produce more or less equal amounts of IFN-beta (Figure 5 and 6). However, when normalized to the number of CFU at 2 hours post infection, the SecA2 KO strain and the complemented strains produce more IFN-beta than wild type. This is contrary to expectations and questions the validity of this approach. The authors should perform infections with a range of MOIs (lower, equal an higher than presented in the manuscript) with the wild type strain and measure CFU 2 hours post infection and the INF-beta response. If there is a relation between the CFU 2 hours post infection and the IFN-beta response, the normalization is valid, otherwise it adds noise to the data.

4) The amount of IFN-beta produced by wild type marinum is equal to the amount produced by a Sec2A tuberculosis knock out (ref 11 as mentioned in manuscript). I would therefore not expect to see reduced amounts of IFN-beta in a SecA2 knockout. Comparing from one manuscript to an article can be problematic (even though results are from the same laboratory). To put the presented results in better context, the authors should perform experiments with TB in parallel to directly compare the amount of IFN beta produced (WT and SecA2 mutant, M. marinum and M. tuberculosis). These experiment are not required if an earlier time point does show a clear difference

Minor issues

Figure 5: normalization to 10000 bacteria. In subfigure a there is a ~2 fold difference between wild-type and delta SecA2. In subfigure b there is a ~10% difference between wild-type and delta SecA2. However, when normalized (b divide by a) there is a 4 fold difference (subfigure c). This cannot be correct based on the data presented. This should be corrected.

C is not described in legend

Figure 6: C is not described in legend

227 intubated

382 island

Reviewer #2: The authors in this manuscript make an effort to evaluate if SecA2 of M. marinum is essential for generation of a RigI-dependent MAVS signalling. Though SecA2 is conserved across M. marinum and Mtb, they found SecA2-dependent activation of host Rig-I/MAVS cytosolic sensors and subsequent induction of IFN-β is not conserved in M. marinum.

Major comments:

Introduction: At the end of the Introduction, it is important to include a paragraph on which salient approaches were employed in the current study against which objectives and the main outcomes of the study.

Line 94-96: It is unclear if the authors meant to indicate that the current study looks into the dependency of SecA2 for activation of Rig-I in general in mycobacteria or was this already established in other species of mycobacteria (if so, authors to cite appropriate references) and they are currently evaluating if M. marinum also exploits similar mechanism. Rewrite these lines for more clarity.

Figure 2: Shouldn’t the levels of SecA2 in complemented be close to the levels of the WT? Wouldn’t the very high levels of SecA2 protein influence the outcome of Rig-I/MAVS activation? Authors to comment in detail. Additionally, explain (i) why there are two SecA2 bands in the complemented strains and not in the WT and �esxBA; (ii) Why the bands of SecA2 of MT and MM are of different size given the very small difference in their total lengths? and (iii) Why the SecA2 of MM is marked lower and SecA2 of Mt is marked higher?

Can the authors comment on the quantity of RNA released with and without SecA2 by MT and MM? Is the IFN-β solely dependent on the release of RNAs?

Given the results, it would have been best if the authors would have generated a secA2 KO in MT and complemented it with both the MM- and MT-encoded SecA2. However this was not performed. Can the authors comment on why they thought this was unnecessary especially after their observations?

Minor comments:

Line 80 and others: Change all through the manuscript - tuberculosis to ‘Tuberculosis’

Line 82, 114, 121, 551, 555, : Unable to see which Interferon! Please change the square box to the appropriate symbol.

Line 229-230: Include the images of deletion confirmation by the genotyping data and Sanger sequencing in Supplementary.

Figure 1: Include an alignment image of SecA2 proteins of MT and MM with exact identity and similarity details.

Line 572-573: When in Fig 3A, �secA2 grows slowly than the rest, how can it still have equal number of bacteria to other strains at 1.0 O.D. Please explain. Isn’t it possible that different number of each strain would have entered into the macrophages at time 0 h? Are the numbers of each strain at 0 h recorded and if so (share in supplementary), similar if not identical?

Line 587-596 & 626-636: Fig 5 and 6 legends lacks explanation for C. Please include

6. PLOS authors have the option to publish the peer review history of their article (what does this mean?). If published, this will include your full peer review and any attached files.

Reviewer #1: No

Reviewer #2: No

---

## [Author Response · Author response to Decision Letter 0]

1 Oct 2023

Please see attached file for version with color coded highlighting.

Reviewer #1: The authors aim to show that the SecA2 secretion system of mycobacterium marinum does contribute to IFN-beta production. This would make for a more convenient model on BSL-2 to study this process in mycobacteria. Unfortunately no SecA2 depended type I interferon production was observed in M. marinum. Even replacement of the SecA2 secretion system of marinum with the mycobacterium tuberculosis version, does not result in SecA2 dependent IFN-beta production. Furthermore the authors correlate the amount of IFN-beta produce to the CFU at time 2hous post infection to relate IFN-beta production to the number of bacteria to circumvent any bias. This could have some additional value, since uptake and e.g. clumping of the bacteria can all affect the CFU recovered at 2h post infection and therefore the response measured. The validity of the normalization is however uncertain. The authors present negative data, that could be interesting, but should provide a more evidence to support their claims.

Major Concerns:

1) The functionality of the complementation strains is tested to a limited extend, especially for the complementation with the SecA2 from tuberculosis. In figure 3B complementation of Sec2A from tuberculosis is missing and should be tested. As an alternative the secretion of e.g. PknG could be tested.

To provide additional data on the functionality of the M. tuberculosis cross-complementation strain, we added a supplemental figure (supplemental figure 7) that examines the ability of the M. marinum ΔsecA2/psecA2MT to grow in the presence of SDS. This data, shows partial complementation for both the M. marinum and M. tuberculosis complementation constructs.

These new results were also addressed in text as described below:

Two such observations of M. marinum ΔsecA2-related growth deficiencies involved increased susceptibility to antibiotics specifically targeting the cell wall (43) and sensitivity to SDS (29), both of which suggest a cell wall synthesis defect. To examine if this phenotype was present in our ΔsecA2 strain, we tested it for sensitivity to 0.05 - 0.2% SDS. In support of previous work by Watkins et al., our ΔsecA2 strain also showed an observable growth defect in SDS (0.052 relative growth +/- 0.037 s.d.; Figure 3B) as compared to growth in the same media lacking SDS. This phenotype was partially or fully rescued to WT levels (1.61 relative growth +/- 1.17 s.d.) by complementation with the M. marinum or M. tuberculosis version of secA2 (1.88 relative growth +/- 0.98 s.d.; Figure 3B-C; Supplemental Figure 7). Similar to growth in nutrient rich media, the degree of complementation varied across different media preparations.

2) Time of measurement at 24h. M. marinum is a more fast paced pathogen. It grows faster, but also escapes (has access to) earlier to the cytosol. In reference 11 (as mentioned in manuscript) in figure 3 it is clear that the IFN-beta response diminishes over time. Measuring at 24 hours might be too late to observe any difference in response. It is therefore imperative to measure earlier, e.g. 6-8 hours post infection. This can be done by either ELISA but also qPCR.

Reviewer 1 makes an important point that the 24 hour time point used in an M. tuberculosis infection model may not be appropriate when measuring the response to the faster growing M. marinum mycobacterial species. To address this concern, we have provided additional data (please see Supplemental figure 12 and associated explanation included below) which shows the ΔsecA2 M. marinum strain induces levels IFN-β, Irf7, and Rig-I comparable to WT. This is consistent with 24-hour ELISA data. Notably for both pieces of data, as expected, macrophages infected with the ΔesxBA M. marinum showed lower levels of IFN-β compared to macrophages infected with the WT strain. 

“As the kinetics of an M. marinum infection are faster than that of M. tuberculosis, with a faster doubling rate and quicker access to the host cytosol, we also examined the host response to M. marinum infection earlier in the infection by quantitative PCR (Supplemental figure 12). We assessed BMDMs infected with our panel of M. marinum strains for relative levels of transcript abundance for IFN-β, the transcription factor Irf7, and the RNA sensor Rig-I relative to the housekeeping gene GAPDH at 8 hours post infection. Results from this assay showed no difference in transcript abundance for any of the genes tested across all strains except ΔesxBA, where there was a significant reduction in transcript abundance for all three genes as compared to WT (Supplemental figure 12).”

3) Normalization strategy. All strains except for the EsxBA mutant produce more or less equal amounts of IFN-beta (Figure 5 and 6). However, when normalized to the number of CFU at 2 hours post infection, the SecA2 KO strain and the complemented strains produce more IFN-beta than wild type. This is contrary to expectations and questions the validity of this approach. The authors should perform infections with a range of MOIs (lower, equal an higher than presented in the manuscript) with the wild type strain and measure CFU 2 hours post infection and the INF-beta response. If there is a relation between the CFU 2 hours post infection and the IFN-beta response, the normalization is valid, otherwise it adds noise to the data.

To address Reviewer 1’s concern about the normalization strategy, we performed M. marinum infections with a range of MOIs for the ΔsecA2 strain (1, 2.5, 5, and 10). For each infection, we also measured the number of bacteria present 2hpi and conducted the normalization as previously described for Figures 5 and 6. For each biological replicate, our data shows a linear relationship between the number of CFUs and IFN-β response. This data is now included as Supplemental figures 10 and 11 and is described further in the results section with the following text:

“We additionally assessed the impact increasing numbers of ΔsecA2 M. marinum (with MOIs ranging from 1 to 10) had on secreted levels of IFN-β (Supplemental figure 10 A-C). Higher CFU counts at 2hpi induced a strong linear increase in the IFN-β in response for each biological replicate, although with variation across replicates in terms of the magnitude of the response elicited (slope of the line; Supplemental figure 11). Notably, when normalized, ΔsecA2 M. marinum induced an average amount of IFN-β around 7 times higher than WT for all MOIs tested (Supplemental figure 10C).”

4) The amount of IFN-beta produced by wild type marinum is equal to the amount produced by a Sec2A tuberculosis knock out (ref 11 as mentioned in manuscript). I would therefore not expect to see reduced amounts of IFN-beta in a SecA2 knockout. Comparing from one manuscript to an article can be problematic (even though results are from the same laboratory). To put the presented results in better context, the authors should perform experiments with TB in parallel to directly compare the amount of IFN beta produced (WT and SecA2 mutant, M. marinum and M. tuberculosis). These experiment are not required if an earlier time point does show a clear difference

While we understand Reviewer 1’s concern about differences in secreted levels of IFN-β following infection with M. tuberculosis as compared with M. marinum we do not believe a side-by-side experiment would provide a deeper understanding of the conservation of secA2 mediated pathogeneses between the two species. Because of differences between the species, including: replication rate, optimal growth temperature, multiplicity of infection (due to M. marinum’s ability to cause macrophage cytolysis, for example, we infect at a much lower MOI), etc. we do not expect to necessarily observe comparable concentrations of secreted IFN-β from BMDMs as a result of M. marinum versus M. tuberculosis infections. Our aim was not to directly compare levels of IFN-β induced following infection of M. marinum or M. tuberculosis but to examine if secA2 contributed to IFN-β mediated pathogenesis as it does in M. tuberculosis. Based on previous literature, we would expect to see a significant difference in IFN-β concentration following infection with a WT versus ΔesxBA (an attenuated strain known to induce low levels of IFN-β). Therefore, we included this control in all of our infection experiments to ensure there was an adequate dynamic range to evaluate the IFN-β response following infection. If a secA2 mediated type I interferon response were conserved between M. marinum and M. tuberculosis, we would expect to see levels of IFN-β closer to the ΔesxBA response than to the WT. However, our data suggests that the ΔsecA2 strain induces levels of IFN-β equal to or higher than WT rather than lower, suggesting this pathogenesis mechanism is not conserved.

In addressing the first major concern from Reviewer 2 about the final paragraph of the introduction, we included an additional paragraph that more clearly states the aim of the paper. This is also partially addressed in the response provided to comment 2 of Reviewer 1.

Minor issues

Figure 5: normalization to 10000 bacteria. In subfigure a there is a ~2 fold difference between wild-type and delta SecA2. In subfigure b there is a ~10% difference between wild-type and delta SecA2. However, when normalized (b divide by a) there is a 4 fold difference (subfigure c). This cannot be correct based on the data presented. This should be corrected.

Reviewer 1 brings up an important point here. We have double checked the math corresponding to the normalized data presented in subfigure C of figures 5 and 6. The roughly 4 fold difference observed is correct as it takes into account the average value of the ratio of IFN-β response to number of CFUs present at 2hpi for each biological replicate individually (∑(b/a) / n). If instead we chose to take the ratio of averages (ratio of the average IFN-β value for all replicates over the average CFU value for all replicates; ∑b/∑a) we would instead get a ~1.5 fold change. While this is also a valid method to examine the data, we felt it was less appropriate in this case because it first averages all 10 biological replicates into one data point rather than averaging individual ratios for each replicate. We have now explained this in a subsection of Materials and Methods (starting at line 398): 

“IFN-β normalization 

Secreted levels of IFN-β were normalized to 10,000 bacteria using the equation a/∑b, where “a” is the amount of IFN-β measured for each technical replicate and “∑b” is the sum of CFUs counted from technical replicates originating from matching wells of the same biological experiment. This number was then multiplied by 10,000 to provide IFN-β values normalized to a standard number of bacteria present in the host cell.”

C is not described in legend

Figure 6: C is not described in legend

We have now added the below text into the Figure 5 and 6 legends to describe panel C of those figures.

“To account for variability in the number of bacteria present at 2hpi, IFN-β levels were normalized to a standard number of bacteria (C).”

227 intubated

Intubated was changed to incubated at line 227.

382 island

Island was replaced with and at line 382.

Reviewer #2: The authors in this manuscript make an effort to evaluate if SecA2 of M. marinum is essential for generation of a RigI-dependent MAVS signalling. Though SecA2 is conserved across M. marinum and Mtb, they found SecA2-dependent activation of host Rig-I/MAVS cytosolic sensors and subsequent induction of IFN-β is not conserved in M. marinum.

Major comments:

Introduction: At the end of the Introduction, it is important to include a paragraph on which salient approaches were employed in the current study against which objectives and the main outcomes of the study.

We understand that some introduction sections include a final paragraph describing the approaches employed in the current study and main outcomes. We have included an additional paragraph at the end of the introduction that briefly outlines the salient approaches employed in the current study (please see below) but omit any discussion of the main outcomes as we prefer to leave this information for the results and discussion section. 

“To test this hypothesis, we generated a ΔsecA2 M. marinum strain as well as M. marinum complementation and M. tuberculosis cross complementation constructs. Using these strains, we examined the impact secA2 has on bacterial growth and virulence under a variety of different in vitro and ex vivo conditions. To specifically address our primary aim of examining whether the SecA2 protein secretion system contributes to an RNA driven, IFN-β mediated pathogenesis mechanism in M. marinum as it does in M. tuberculosis, we infected WT and MAVS-/- BMDMs with our generated M. marinum strains and examined the macrophage secretome for secreted levels of IFN-β.”

Line 94-96: It is unclear if the authors meant to indicate that the current study looks into the dependency of SecA2 for activation of Rig-I in general in mycobacteria or was this already established in other species of mycobacteria (if so, authors to cite appropriate references) and they are currently evaluating if M. marinum also exploits similar mechanism. Rewrite these lines for more clarity.

Lines 94-96 are included as a means to transition from what we know about Rig-I activation by M. tuberculosis to describe the role the SecA2 protein secretion system plays in this response as well as M. tuberculosis pathogenesis more broadly. We made the changes below to further emphasize that this characterisation is being described for M. tuberculosis. 

“In M. tuberculosis, there are multiple mechanisms by which IFN-β is induced following infection (20–24), including through the activation of host cytosolic RNA sensors canonically described for their role in driving the type I interferon response to viruses (11,25,26). Activation of these host RNA sensors has been shown to be driven by the release of bacterially derived RNAs into the host cytosol (11). Once within the cytosol, these immunostimulatory RNAs are recognized by retinoic acid-inducible gene I (RIG-I), causing a conformational change in this protein which enables it to interact with the signal transducing adaptor protein, mitochondrial antiviral-signalling protein (MAVS). Through a series of phosphorylation-dependent steps, this signal is ultimately relayed to transcription factors such as interferon regulatory factor 3 (Irf3) and Irf7, which translocate into the nucleus to drive IFN-β transcription (27). The mechanism by which these M. tuberculosis-derived RNAs gain access to the host cytosol is unclear, but it is known to be dependent, in part, on a functional SecA2-protein secretion system.”

Figure 2: Shouldn’t the levels of SecA2 in complemented be close to the levels of the WT? Wouldn’t the very high levels of SecA2 protein influence the outcome of Rig-I/MAVS activation? Authors to comment in detail. Additionally, explain (i) why there are two SecA2 bands in the complemented strains and not in the WT and esxBA; (ii) Why the bands of SecA2 of MT and MM are of different size given the very small difference in their total lengths? and (iii) Why the SecA2 of MM is marked lower and SecA2 of Mt is marked higher?

Shouldn’t the levels of SecA2 in complemented be close to the levels of the WT? 

We agree with reviewer 2 that ideally the complementation strains would show equal levels of secA2 expression. However, this result is not unexpected as secA2 in the complementation strain is under the control of the MOPS promotor rather than its native promoter. We have added text to the appropriate section of the results to provide additional clarification, please see below. 

“The higher levels of secA2 transcript abundance observed in the complement and cross-compliment strains are likely due to constitutive expression by the mycobacterial optimal promoter (MOP), rather than its native promoter.”

- why there are two SecA2 bands in the complemented strains and not in the WT and desxBA.

We are unsure as two what the bands above and below SecA2 for the M. marinum and M. tuberculosis on the western blot are. We share Reviewer 2’s curiosity of what they might be and why they only appear in those two strains but without further analysis this is unclear. As they do not seem to impact complementation, we have not explored them further, although we did add additional text to address this concern:

“Our results indicate a roughly 90 kDa protein present in all strains except the ΔsecA2 (Figure 2F), indicated by the red arrow to differentiate it from lower and higher molecular weight species in the M. marinum and M. tuberculosis complementation strains. While we are unsure what these additional protein bands represent, they may be the result of post-translational modifications or degradation products of the SecA2 protein, as they are missing from the ΔsecA2 strain. Taken together, our results confirm the generation of an unmarked ΔsecA2 deletion as well as M. marinum and M. tuberculosis complementation and cross-complementation constructs.”

- Why the bands of SecA2 of MT and MM are of different size given the very small difference in their total lengths?

SecA2 is roughly the same size for all strains that express it, as indicated by the red arrow on the western blot for figure 3. There are slight differences in abundance of the protein, which might contribute to the sizes looking slightly different. We have added text (please see above) to make this clearer.

-Why the SecA2 of MM is marked lower and SecA2 of Mt is marked higher? 

This is a great question. We do not know what is causing the lower and higher molecular weight protein products. It is possible they are caused by post-translational modifications or degradation of the protein. Further analysis of the proteins would need to be conducted to define whether these bands result from post-translational modification or are caused by SecA2 degradation. As the additional bands do not seem to impact complementation (they phenotypically behave like WT), we believe further analysis of the protein is outside the scope of this paper. We have added text (please see above) to make this clearer.

Can the authors comment on the quantity of RNA released with and without SecA2 by MT and MM? Is the IFN-β solely dependent on the release of RNAs?

To date, the quantity of RNA released with and without SecA2 by M. tuberculosis has not been published. It is not known how much RNA is released with/without SecA2 by M. marinum. Had we observed a SecA2-dependent induction of the Rig-I/MAVS signalling complex, we would have likely pursued a deeper exploration of the quantity and characteristics of RNAs being released in a SecA2 dependent manner. No, there are many different mechanisms though which IFN-β is induced, including by DNA released in an ESX-1 dependent manner.

Given the results, it would have been best if the authors would have generated a secA2 KO in MT and complemented it with both the MM- and MT-encoded SecA2. However, this was not performed. Can the authors comment on why they thought this was unnecessary especially after their observations?

We agree with reviewer 2 that generating an M. marinum and M. tuberculosis complementation strain in an M. tuberculosis ΔsecA2 background would have provided additional insight into the conservation of this unique pathogenesis mechanism. Had we observed a conservation is SecA2-mediated induction of a type I interferon response following infection of BMDMs with M. marinum, this would have been an experiment we would have likely pursued next.

Minor comments:

Line 80 and others: Change all through the manuscript - tuberculosis to ‘Tuberculosis’

Thank you for your comment. We left all instances of tuberculosis with a lower case “t” to better adhere to scientific naming convention and because the disease name does not require capitalization.

Line 82, 114, 121, 551, 555, : Unable to see which Interferon! Please change the square box to the appropriate symbol.

Thank you for pointing this out. We have re-inserted the symbol where described.

Line 229-230: Include the images of deletion confirmation by the genotyping data and Sanger sequencing in Supplementary.

A supplemental figure (Supplemental figure 2) has been added with the Sanger sequence chromatogram for the ΔsecA2 strain. Genotyping data is presented in Figure 2A-D.

Figure 1: Include an alignment image of SecA2 proteins of MT and MM with exact identity and similarity details.

An alignment of the amino acid sequences for M. marinum and M. tuberculosis SecA2 is included in Supplemental figure 1. Conservation of amino acid sequences is denoted by an asterix (*).

Line 572-573: When in Fig 3A, secA2 grows slowly than the rest, how can it still have equal number of bacteria to other strains at 1.0 O.D. Please explain. Isn’t it possible that different number of each strain would have entered into the macrophages at time 0 h? Are the numbers of each strain at 0 h recorded and if so (share in supplementary), similar if not identical?

Reviewer 2 is correct that is is possible that a different number of bacteria from each strain could be present in the macrophage at timepoint 2hpi (at the end of the infection period). We shared this concern, which is why we collected data on the number of bacteria present in BMDMs at the end of the infection period (2hpi). This data is provided in figure 4B as well as panel A of Figures 5 and 6. We observed variation in the number of bacteria present at 2hpi across strains, which prompted us to introduce a normalization technique. This is described in lines 609-610 and 652-654. We also noted that while there were differences in the number of bacteria being taken up, this did not seem to be due to changes in the macrophages ability to phagocytose the bacteria (Supplementary figure 9) but more likely due to the equation applied to estimate the number of bacteria added.

Line 587-596 & 626-636: Fig 5 and 6 legends lacks explanation for C. Please include

We have now added the below text into the Figure 5 and 6 legends to describe panel C of those figures.

“To account for variability in the number of bacteria present at 2hpi, IFN-β levels were normalized to a standard number of bacteria (C).”

---

## [Decision Letter · Decision Letter 1]

3 Nov 2023

Mycobacterium tuberculosis SecA2-dependent activation of host Rig-I/MAVs signaling is not conserved in Mycobacterium marinum

PONE-D-23-02235R1

Dear Dr. Serene,

We’re pleased to inform you that your manuscript has been judged scientifically suitable for publication and will be formally accepted for publication once it meets all outstanding technical requirements.

Kind regards,

Atul Vashist, PhD

Academic Editor

PLOS ONE

Additional Editor Comments (optional):

Reviewers' comments:

Reviewer's Responses to Questions

**Comments to the Author**

1. If the authors have adequately addressed your comments raised in a previous round of review and you feel that this manuscript is now acceptable for publication, you may indicate that here to bypass the “Comments to the Author” section, enter your conflict of interest statement in the “Confidential to Editor” section, and submit your "Accept" recommendation.

Reviewer #2: All comments have been addressed

Reviewer #3: All comments have been addressed

2. Is the manuscript technically sound, and do the data support the conclusions?

Reviewer #2: Yes

Reviewer #3: Yes

3. Has the statistical analysis been performed appropriately and rigorously? 

Reviewer #2: Yes

Reviewer #3: Yes

4. Have the authors made all data underlying the findings in their manuscript fully available?

Reviewer #2: Yes

Reviewer #3: Yes

5. Is the manuscript presented in an intelligible fashion and written in standard English?

Reviewer #2: Yes

Reviewer #3: Yes

6. Review Comments to the Author

Reviewer #2: The authors have taken efforts to address all the queries and responded appropriately. All experimental recommendations from reviewers has been addressed. While they are at loss to provide approproiate reasons for altered MM and MT SecA2 protein sizes, the provided modifed manuscript fullfills the requirements for manuscript acceptance

Reviewer #3: The study discuss the important aspect of SecA protein in Mycobacteria and this has some relevance in the pathogenesis. However it is advised that author should continue this study and demonstrate in vivo and clinical relevance of the concept in broader perspective in their future study

7. PLOS authors have the option to publish the peer review history of their article (what does this mean?). If published, this will include your full peer review and any attached files.

Reviewer #2: No

Reviewer #3: No

---

## [Editor Report · Acceptance letter]

12 Feb 2024

PONE-D-23-02235R1 

PLOS ONE

Dear Dr. Serene, 

I'm pleased to inform you that your manuscript has been deemed suitable for publication in PLOS ONE. Congratulations! Your manuscript is now being handed over to our production team.

Kind regards, 

on behalf of

Dr. Atul Vashist 

Academic Editor

PLOS ONE